# Be Careful What You Smooth For: Label Smoothing Can Be a Privacy Shield but Also a Catalyst for Model Inversion Attacks

**Lukas Struppek**
Technical University of Darmstadt
German Center for Artificial Intelligence
struppek@cs.tu-darmstadt.de

**Dominik Hintersdorf**
Technical University of Darmstadt
German Center for Artificial Intelligence
hintersdorf@cs.tu-darmstadt.de

**Kristian Kersting**
Technical University of Darmstadt
Centre for Cognitive Science of TU Darmstadt
Hessian Center for AI (hessian.AI)
German Center for Artificial Intelligence

Label smoothing – using softened labels instead of hard ones – is a widely adopted regularization method for deep learning, showing diverse benefits such as enhanced generalization and calibration. Its implications for preserving model privacy, however, have remained unexplored. To fill this gap, we investigate the impact of label smoothing on model inversion attacks (MIAs), which aim to generate class-representative samples by exploiting the knowledge encoded in a classifier, thereby inferring sensitive information about its training data. Through extensive analyses, we uncover that traditional label smoothing fosters MIAs, thereby increasing a model's privacy leakage. Even more, we reveal that smoothing with negative factors counters this trend, impeding the extraction of class-related information and leading to privacy preservation, beating state-of-the-art defenses. This establishes a practical and powerful novel way for enhancing model resilience against MIAs.
Source code: https://github.com/LukasStruppek/Plug-and-Play-Attacks

## 1 Introduction

Deep learning classifiers continue to achieve remarkable performance across a wide spectrum of domains (Radford et al., 2021; Ramesh et al., 2022; OpenAI, 2023), due in part to powerful regularization techniques. The common Label Smoothing (LS) regularization (Szegedy et al., 2016) replaces labels with a smoothed version by mixing the hard labels with a uniform distribution to improve generalization and model calibration (Pereyra et al., 2017; Müller et al., 2019). However, the very capabilities that make these models astonishing also render them susceptible to privacy attacks, potentially resulting in the leakage of sensitive information about their training data.

One category of privacy breaches arises from model inversion attacks (MIAs) (Fredrikson et al., 2015), a class of attacks designed to extract characteristic visual features from a trained classifier about individual classes from its training data. In the commonly investigated setting of face recognition, the target model is trained on facial images to predict a person's identity. Without any further information about the individual identities, MIAs exploit the target model's learned knowledge to create synthetic images that reveal the visual characteristics of specific classes. As a practical example, let us take a high-security facility that uses a face recognition model for access control. MIAs could enable unauthorized adversaries to reconstruct facial features by accessing the face recognition model without any further information required and with the goal of inferring the identity of authorized staff. In this case, a successful attack can lead to access control breaches and potential security and privacy threats to individuals.

While recent MIA literature (Zhang et al., 2020; Struppek et al., 2022a) keeps pushing the attacks' performance, the influence of the target model's training procedure on the attack success has not been studied so far. We are the first to start investigations in this direction with a focus on LS regularization, whose effects on a model's privacy have not yet been considered. We connect LS to a model's privacy leakage in the light of MIAs and reveal that training with positive, i.e., standard label smoothing

increases a model's vulnerability, particularly in low data regimes. Moreover, we reveal that by smoothing labels with a negative smoothing factor and, therefore, penalizing a model's confidence in any class different from the true label, one can make models robust to MIAs and decrease their privacy leakage significantly without large performance drops on its classification tasks, offering a better utility-privacy trade-off than existing defense approaches. Our extensive evaluation studies the various effects of both positive and negative label smoothing on MIAs and explores possible reasons for these phenomena. These insights not only contribute to our understanding of model privacy but also bear practical implications for deep learning applications. By strategically deploying LS, model providers can balance model performance with security and mitigate the risks associated with privacy breaches in various real-world scenarios.

In summary, we make the following contributions:

- We are the first to demonstrate that positive label smoothing increases a model's privacy leakage in light of model inversion attacks, particularly in low data regimes.
- We reveal that negative label smoothing counteracts this effect and offers a practical defense, beating state-of-the-art approaches with a better utility-privacy trade-off.
- We introduce a novel attack metric and provide rigorous investigations on the effects of label smoothing on the target models and the individual stages of model inversion attacks.

## 2 BACKGROUND AND RELATED WORK

We start by introducing model inversion attacks (Sec. 2.1) and label smoothing (Sec. 2.2).

### 2.1 MODEL INVERSION ATTACKS

In the image classification setting, let $\mathcal{M}_{target} \colon X \to [0,1]^C$ be a classifier that takes images $x \in X$ and computes for each class $c \in \{1, \ldots, C\}$ a probability $p_c \in [0,1]$. Model inversion attacks (MIAs) aim to construct synthetic images that reflect and reveal characteristic features of a specific class $c$ learned by $\mathcal{M}_{target}$. In the standard MIA setting, the adversary has only access to $\mathcal{M}_{target}$ and knows the general data domain but has no detailed information about the individual classes. In the face recognition domain, a common setting for MIAs, each class corresponds to an individual identity, but the names and appearances of these identities are unknown to the adversary. A successful MIA allows for inferring the visual appearance and identity of the different classes from the training data without direct data access, leading to a notable security breach.

The first MIAs were limited to linear regression (Fredrikson et al., 2014) and shallow neural networks (Fredrikson et al., 2015) by using gradient-based sample optimization to reveal features. Direct sample optimization is prone to producing adversarial examples (Szegedy et al., 2014), i.e., inputs that look nothing like the target class but are still assigned high prediction scores by the target model. To overcome this problem, Zhang et al. (2020) added a generative adversarial network (GAN) (Goodfellow et al., 2014) as prior to enable attacks against deeper networks and improve the image quality of the generated images. GANs consist of two components, a generator network $G \colon Z \to X_{prior}$, which is trained to generate images $x \in X_{prior}$ from latent vectors $z \in Z$, and a discriminator network $D$ used to distinguish between generated images and real images. Generative MIAs then try to solve the following optimization goal by optimizing a latent vector $\hat{z}$:

$$\min_{\hat{z}} \mathcal{L}(\mathcal{M}_{target}, G, D, \hat{z}, c) \,. \tag{1}$$

Here, $\mathcal{L}$ denotes a suitable loss function to maximize the target model's confidence in the target class $c$, e.g., a cross-entropy loss computed on the outputs of $\mathcal{M}_{target}$ for images generated by $G$. Broadly speaking, the adversary tries to find a spot on the generative model's manifold that represents the visual features of the target class and, ideally, allows inferring the person's identity. To find such a spot, the target model's knowledge is exploited to provide guidance through the latent space. This is not a trivial task since MIAs face various challenges, including misleading feature reconstruction, distributional shifts, and complex optimization landscapes (Struppek et al., 2022a). To tackle these challenges, generative MIAs have been improved, e.g., by training target-specific GANs (Chen et al., 2021; Yuan et al., 2023) or changing the attack's objective function (Wang et al., 2021a; Nguyen et al., 2023). Whereas most MIAs require white-box model access, some gradient-free approaches

were also proposed. Han et al. (2023) introduced a reinforcement learning-based black-box attack, and Kahla et al. (2022) and Zhu et al. (2023) proposed label-only MIAs.

However, all mentioned attacks focus on low-resolution tasks and have yet to prove successful in high-resolution data regimes. Recently, Struppek et al. (2022a) introduced Plug & Play Attacks (PPA), which decoupled the attack and target model from the underlying GAN, allowing the use of any suitable pre-trained GANs from the target domain. The attack has demonstrated greater robustness to distributional shifts, increased flexibility in the choice of generative prior and target model, and high effectiveness in the high-resolution setting. Therefore, we base most of our experiments on PPA since it allows us to investigate a more realistic attack scenario with high-resolution data. We provide a more detailed overview of PPA's specifics in Appx. B.4.

## 2.2 LABEL SMOOTHING REGULARIZATION

Image classifiers are typically trained using a cross-entropy loss $\mathcal{L}_{CE}(\mathbf{y}, \mathbf{p}) = -\sum_{k=1}^{C} y_k \log p_k$, where the ground-truth class label vector $\mathbf{y}$ assigns a value of 1 to the correct class $c$ and sets all other values to zero. Here, $\mathbf{p} \in [0,1]^C$ represents the model's output probability vector for the current sample. In the standard hard label setting, each training sample is strictly assigned to a single class $c$, simplifying the loss to $\mathcal{L}_{CE} = -\log p_c$. During training, the model learns to produce high values for the predicted class with $p_c \gg p_k$ and $c \neq k$, which often results in overconfident models. To mitigate this effect, Szegedy et al. (2016) introduced label smoothing (LS) regularization. LS replaces the hard label with a mixture of the hard-coded label and a uniformly distributed vector. Formally, the target vector $\mathbf{y}^{\text{LS}}$ for positive LS with a smoothing factor $\alpha \in (0,1]$ and $C$ classes is defined by

$$\mathbf{y}^{\text{LS}} = (1 - \alpha) \cdot \mathbf{y} + \frac{\alpha}{C} \,. \tag{2}$$

For example, let $\mathbf{y} = (1, 0, 0)^T$ be a one-hot encoded target vector. Smoothing with $\alpha = 0.3$ replaces the vector by $\mathbf{y}_{pos}^{\text{LS}} = (0.8, 0.1, 0.1)^T$. This target vector represents uncertainty about the true label and encodes the correlation of the sample to classes different from the assigned hard label. In combination with $\mathcal{L}_{CE}$, LS effectively replaces $\mathcal{L}_{CE}$ loss with a weighted combination of losses:

$$\mathcal{L}^{\text{LS}}(\mathbf{y}, \mathbf{p}, \alpha) = (1 - \alpha) \cdot \mathcal{L}_{CE}(\mathbf{y}, \mathbf{p}) + \frac{\alpha}{C} \cdot \sum_{k=1}^{C} \mathcal{L}_{CE}(\mathbf{1}, \mathbf{p}) \,. \tag{3}$$

Here, $\mathbf{1}$ denotes a vector of length $C$ with all entries set to 1. Previous research has shown that LS improves generalization (Szegedy et al., 2016; Pereyra et al., 2017), model calibration (Müller et al., 2019), language modeling (Chorowski & Jaitly, 2017), and learning in low label noise regimes (Lukasik et al., 2020). Wei et al. (2022) generalized the LS formulation by allowing $\alpha \in (-\infty, 1]$ and demonstrated that smoothing with a negative factor can improve model performance in high label noise regimes. Smoothing with a negative factor creates target vectors that are no longer valid probability distributions. Repeating the previous example with a negative smoothing factor $\alpha = -0.3$ results in the target vector $\mathbf{y}_{neg}^{\text{LS}} = (1.2, -0.1, -0.1)^T$. Training with negative LS not only encourages the model to learn the target class but also penalizes confidence in other classes. This can formally be seen in Eq. (3), where the second term becomes negative. A more formal analysis of LS is provided in Appx. A. While LS demonstrated performance improvements in various domains (Szegedy et al., 2016; Zoph et al., 2018; He et al., 2019), it is important to highlight that LS also increases a model's privacy leakage, an aspect that has not been investigated yet.

## 3 ILLUSTRATING THE DUAL USE OF LABEL SMOOTHING FOR MIAS

To motivate our investigation into the effects of LS on model privacy, we begin with a simple toy example based on a two-dimensional dataset with three classes: *blue squares*, *green circles*, and *orange pentagons*. We first illustrate the impact of positive LS. For this, we trained a three-layer neural network with both hard labels ($\alpha = 0$) and soft labels ($\alpha = 0.05$). In Fig. 1, we visualize the decision boundaries and model confidences over the input space. The model trained on hard labels (Fig. 1a) assigns high-confidence predictions to most inputs, even if they are far away from the decision boundary. In contrast, the model trained with positive LS (Fig. 1b) assigns high confidence only to samples close to the training data. We further simulated a simple MIA by taking a random starting point, here a sample from the class (*green circles*), and optimizing this sample to maximize

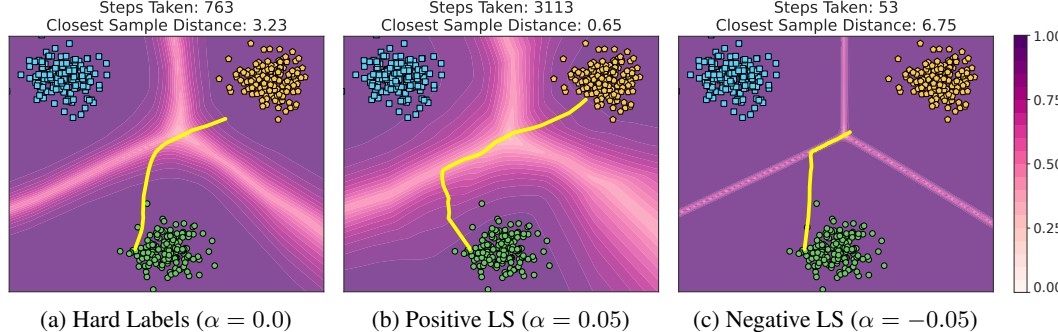

(a) Hard Labels ($\alpha = 0.0$)     (b) Positive LS ($\alpha = 0.05$)     (c) Negative LS ($\alpha = -0.05$)

Figure 1: Simple MIA on a 2D toy dataset with three classes. The Background color indicates the models' prediction confidence, and the yellow lines show the intermediate optimization steps of the attack. The optimization starts from a random position, here a sample from the *green circle* class, and tries to reconstruct a sample from the *orange pentagons* class. The attack against the positive LS model (1b) constructs a sample very close to the targeted training data. In contrast, attacking the negative LS model (1c) saturates close to the decision boundary and far away from the training data.

the model's confidence for another class (*orange pentagons*) to over $95\%$. The goal is to reveal the features of the target class, which are, in this simplified setting, just the coordinates of training samples. While the attack is much faster on the model trained with hard labels – 763 optimization steps compared to 3113 steps – the resulting data point is significantly further away from the training data – 3.23 compared to 0.65 $\ell_2$ distance to the closest training sample. By clustering the training samples in a high-confidence area, positive LS training reveals their position more precisely to the inversion attack and promotes attack results closer to the true target class distribution.

If positive LS improves the attack success, one would expect that negative smoothing counteracts this effect and renders the inversion attack more challenging to execute. Retraining the model with a negative smoothing factor ($\alpha = -0.05$) indeed shows reversed effects (Fig. 1c). The model's confidence is very high everywhere except for the decision boundaries. This leads to an inversion attack that achieves its goal already after 53 steps but ends far away from the training data. Thus, it appears that there exists a trade-off between regularizing a model with LS and the success of MIAs.

## 4    EXPERIMENTAL EVALUATION

With the previous illustration in mind, we now turn to a real-world scenario: face recognition of high-resolution images. Plug & Play Attacks (PPA) (Struppek et al., 2022a) are the current state-of-the-art MIA on which we base our experimental evaluations. Furthermore, we explored the effects of LS on other MIAs in a low-resolution setting in Appx. C.5 to validate our findings. First, we introduce our experimental protocol (Sec. 4.1) before demonstrating the general impact of LS on the attack success (Sec. 4.2), a model's embedding space (Sec. 4.3), and the individual stages of MIAs (Sec. 4.4). For additional experimental details and results, we refer to Appx. B and Appx. C, respectively.

### 4.1    EXPERIMENTAL PROTOCOL

To ensure consistent conditions and avoid confounding factors, we maintain identical training and attack hyperparameters and seeds across different runs, with adjustments solely made to the smoothing factor during target model training. Specifically, the training samples, including their augmentations, remain consistent across runs. For reproducibility, our source code includes configuration files to recreate the results. Additionally, we state all training and attack hyperparameters in Appx. B.

**Datasets:** In line with previous MIA literature, we focus our investigation on the *FaceScrub* (Ng & Winkler, 2014) and *CelebA* (Liu et al., 2015) datasets for facial recognition. FaceScrub comprises images of 530 different identities with equal gender split. While CelebA contains samples of 10,177 identities, we adhere to the standard MIA evaluation protocol (Zhang et al., 2020) and take only the 1,000 identities with the most samples. All images are resized to $224 \times 224$ for model training.

Table 1: PPA attack results against ResNet-152 models trained on FaceScrub and CelebA. Results are compared to state-of-the-art defenses MID and BiDO. While positive LS ($\alpha = 0.1$) amplifies the attacks, negative LS ($\alpha = -0.05$) beats existing defenses in terms of utility-privacy trade-off. Green indicates attack improvements, whereas red shows better defense compared to the baseline.

| | FaceScrub | | | | CelebA | | | |
|---|---|---|---|---|---|---|---|---|
| Model | ↑ Test Acc | ↑ Acc@1 | ↓ $\delta_{face}$ | ↑ $\xi_{train}$ | ↑ Test Acc | ↑ Acc@1 | ↓ $\delta_{face}$ | ↑ $\xi_{train}$ |
| Standard | 94.9% | 94.3% | 0.71 | 61.2% | 87.1% | 81.8% | 0.74 | 59.8% |
| Pos. LS | 97.4% | 95.2% (+0.9) | 0.63 (-0.08) | 71.0% (+9.8) | 95.1% | 92.9% (+11.1) | 0.61 (-0.13) | 66.1% (+6.3) |
| Neg. LS | 91.5% | 14.3% (-80.0) | 1.23 (+0.52) | 16.5% (-44.7) | 83.6% | 26.4% (-55.3) | 1.04 (+0.3) | 7.1% (-52.7) |
| MID | 91.1% | 92.0% (-2.3) | 0.72 (+0.01) | 73.0% (+11.8) | 80.4% | 78.0% (-3.8) | 0.74 (+0.0) | 70.9% (+11.1) |
| BiDO | 87.1% | 45.4% (-48.9) | 0.91 (+0.2) | 59.3% (-1.9) | 79.9% | 63.7% (-18.1) | 0.81 (+0.07) | 60.6% (+0.8) |

**Models:** We trained ResNet-152 (He et al., 2016), DenseNet-121 (Huang et al., 2017), and ResNeXt-50 (Xie et al., 2017) as target models. All results from the main paper are based on ResNet-152 models; results for other architectures are stated in Appx. C. For negative LS, we trained the first epochs without any smoothing and then gradually increased the negative smoothing to stabilize the training and prevent models from getting trapped in poor minima during the initial epochs.

**Attack Parameters:** We used the official implementations of the different attacks to perform MIAs, employing default parameters. This choice ensures that differences in attack performance do not arise from specific parameter selections. Due to the remarkably high time requirements for MIAs, we performed a single attack against each target model. To reduce random influences, we generated a total of 50 samples per target class, which is significantly more than most related research evaluated.

**Metrics**: We employ various metrics consistent with prior research (Zhang et al., 2020; Struppek et al., 2022a) to evaluate the impact of LS. The target models' utility from the user's perspective is quantified by their accuracy on a holdout test set (*Test Acc*). The following metrics quantify the attack success from the adversary's perspective. Additional metrics are provided in Appx. C.

*Attack Accuracy:* To imitate a human evaluator that judges if reconstructed images depict the target class, a separate Inception-v3 (Szegedy et al., 2016) evaluation model is trained on the target model's training data. The attack is then evaluated by computing the proportions of predictions on the synthetic images that match the target class, i.e., the top-1 (*Acc@1*) and top-5 (*Acc@5*) accuracy.

*Feature Distance:* This metric measures the average $\ell_2$ distance $\delta_{face}$ between the reconstructed images and the nearest samples from the target model's training data in the embedding space of a pre-trained FaceNet (Schroff et al., 2015) model, which predicts visual similarity between faces. We also computed the distance $\delta_{eval}$ in the evaluation model's penultimate feature space. Lower distances indicate that the reconstructed samples more closely resemble the training data.

*Knowledge Extraction Score:* For measuring the extracted discriminative information about distinct classes, we introduce a novel metric. Specifically, we train a surrogate ResNet-50 (He et al., 2016) classifier on the synthetic attack results and measure its top-1 classification accuracy $\xi_{train}$ on the target model's original training data. The intuition behind this metric is that the more successful the inversion attack, the better the surrogate model's ability to distinguish between the classes.

## 4.2 THE IMPACT OF LABEL SMOOTHING ON A MODEL'S PRIVACY LEAKAGE

We begin by showcasing the general effects of positive and negative LS on MIAs. Our attack results in Tab. 1 for models trained on the complete FaceScrub and CelebA datasets, respectively, demonstrate that positive LS (second row) indeed amplifies a model's privacy leakage and enables the attacks to extract characteristic class features more closely related to the training data, as indicated by both $\delta_{face}$ and $\xi_{train}$. Moreover, smoothing with a negative factor (third row) substantially diminishes the attacks' success with only a small reduction in a model's test accuracy. Comparing the defensive effect of negative LS to state-of-the-art defenses, MID (Wang et al., 2021c) and BiDO (Peng et al., 2022), even suggests a more favorable utility-defense trade-off, all without requiring architecture adjustments or complex loss functions to be optimized. In the following, we focus our analyses on the FaceScrub ResNet-152 models.

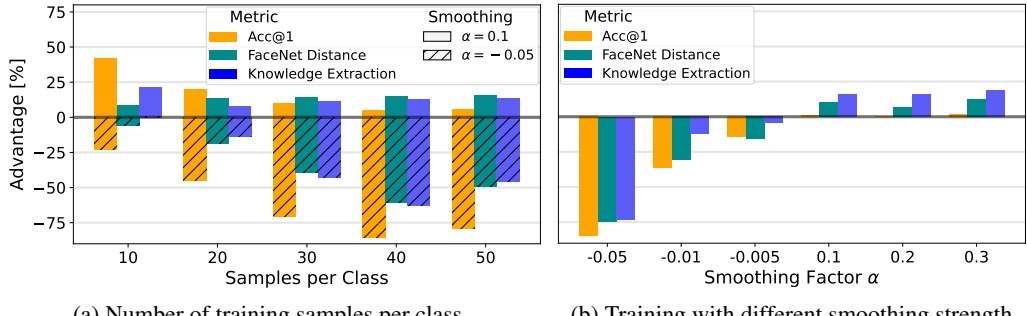

(a) Number of training samples per class    (b) Training with different smoothing strength

Figure 2: Attack results for FaceScrub models trained with varying numbers of training samples per class (2a) and different smoothing factors (2b). Results are stated as the relative improvement, denoted as advantage, compared to the model trained with hard labels. While positive LS has larger impact on low-data regimes, negative LS acts as a stronger defense when trained on more samples.

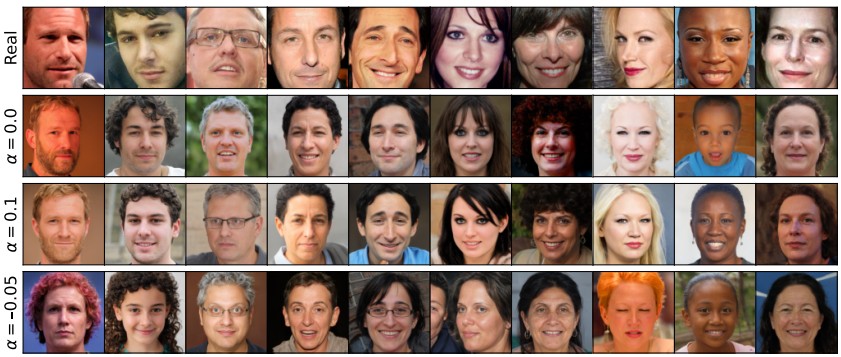

Figure 3: Attack samples from the FaceScrub models trained with 30 samples per class. Samples are not cherry-picked but show the most robust attack results based on PPA's selection procedure. The model trained with positive LS ($\alpha = 0.1$) clearly reveals more visual characteristics of the target identities, whereas attacks on the negative LS model ($\alpha = -0.05$) generate misleading results.

To further investigate the impact of the available number of training samples on the effects of LS, we conducted experiments by training the targets with a fixed number of samples per class to assess whether the effect of LS depends on the training set size. In Fig. 2a, we present the computed metrics as the relative advantage compared to training without LS to make the differences more apparent. Notably, all models trained with positive LS exhibit increased privacy leakage, with the effect being more pronounced in low-data regimes. Conversely, the defensive effects of negative LS relatively improve as the number of training samples increases. The impact of LS is also reflected in the resulting attack samples depicted in Fig. 3. A sensitivity analysis in Fig. 2b further demonstrates that smoothing factors above $\alpha = 0.1$ only marginally contribute to increased privacy leakage, whereas negative factors exhibit an increasingly beneficial defensive effect.

## 4.3 LABEL SMOOTHING'S SHAPING EFFECTS ON EMBEDDING SPACES

The effectiveness of MIAs relies upon a model's ability to discern class-specific features and distinguish them from those of other classes. To assess the influence of LS, we turn our focus to the embedding spaces within the penultimate layer of our ResNet-152 models. These spaces offer lower-dimensional representations of input samples, where inputs considered similar by a model are placed closer together, while dissimilar inputs are placed farther apart. In Fig. 4, we employ t-SNE (van der Maaten & Hinton, 2008) to visualize the embeddings of the derived from training samples across 100 different classes. Without LS (Fig. 4a), the model tends to form clusters among samples from the same class, yet the distinction from other clusters remains rather subtle, with some clusters overlapping. Positive LS (Fig. 4b), however, noticeably enhances the separation between samples from different classes and tightens sample clusters from the same class, which has also been observed

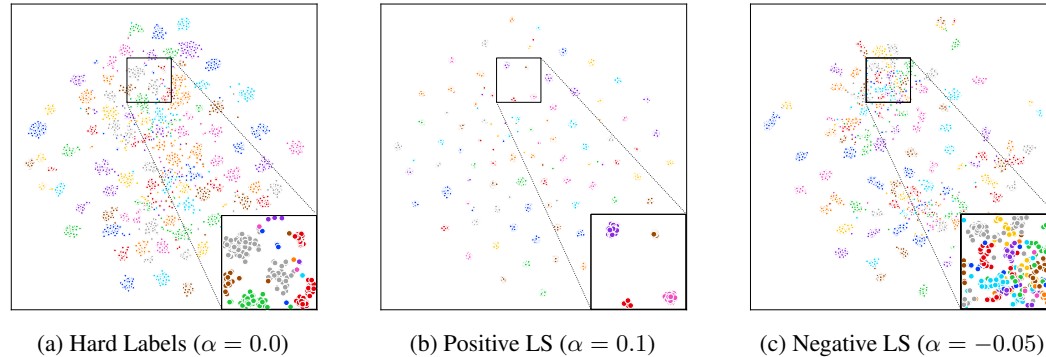

(a) Hard Labels ($\alpha = 0.0$)    (b) Positive LS ($\alpha = 0.1$)    (c) Negative LS ($\alpha = -0.05$)

Figure 4: Penultimate activations of training samples from 100 FaceNet classes (colors are reused). Compared to training with hard labels (4a), training with positive LS (4b) clusters samples from the same class together. Smoothing the labels with a negative factor (4c) reverses this effect and instead places samples from different classes closer together to build a less clearly separated space.

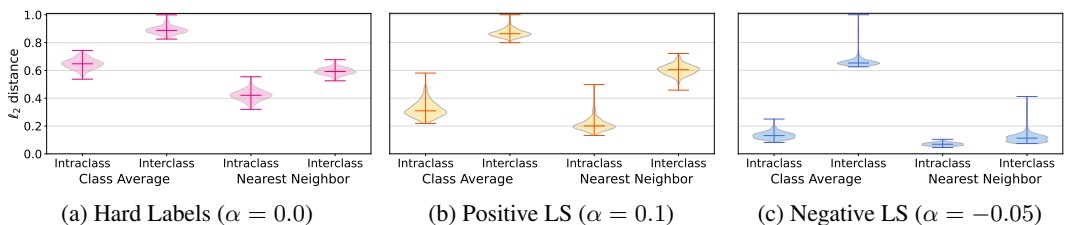

(a) Hard Labels ($\alpha = 0.0$)    (b) Positive LS ($\alpha = 0.1$)    (c) Negative LS ($\alpha = -0.05$)

Figure 5: Distribution of maximum-scaled $\ell_2$ feature distances between penultimate layer activations. The left-hand side of each plot depicts the average distance between each training sample and all other samples from the same class (*intraclass*) and other classes (*interclass*). The right-hand side of each plot shows the distances to the closest sample. Positive LS reduces the relative intraclass distances while increasing the distance to other samples, whereas negative LS partly reverts this effect and moves some samples from other classes closer to samples of a particular class.

by previous research (Müller et al., 2019; Chandrasegaran et al., 2022). This observation suggests that the model has effectively captured discriminative features crucial for identity recognition. On the other hand, training with negative LS (Fig. 4c) partially counteracts this effect, as it promotes increased overlap among different clusters, thereby undermining the clarity of separation.

To establish a quantitative foundation for our observations, we computed the $\ell_2$ distances between the embeddings of all training samples. The violin plots presented in Fig. 5 depict the average distances between samples within the same class (*intraclass*) and all samples from other classes (*interclass*). Additionally, we calculated the average distance from each sample to its nearest neighbor. These results confirm the trends observed in our embedding space visualizations. Specifically, in comparison to training with hard labels (Fig. 5a), the introduction of positive LS (Fig. 5b) effectively diminishes the relative distance between samples belonging to the same class, while simultaneously increasing the separation from samples of other classes. Training with negative LS (Fig. 5c) also increases the relative distances between samples of different classes, but the nearest neighbor interclass distances are comparable to the average interclass distance. This observation suggests that samples within a single cluster exhibit a higher degree of label inconsistency.

In a broader perspective, generative MIAs can be described as the process of exploring the data manifold of the generative model, guided by the target model, to find meaningful representations of specific classes. In this regard, positive LS is expected to enhance the exploration by offering better guidance since the target model is able to better distinguish between the features of different classes. The guidance signal of models trained with negative LS, in turn, contains less clear information, as class embeddings overlap, obfuscating the characteristic features of individual classes. We delve deeper into the impact of LS on the various stages of MIAs in the following section.

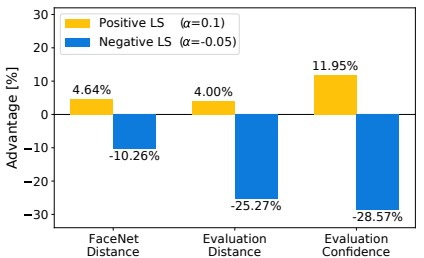
(a) Quality of initially selected vectors

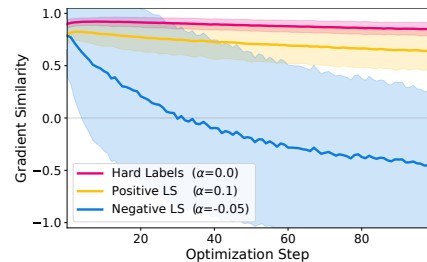
(b) Similarity between consecutive image gradients

Figure 6: Fig. 6a shows the advantage of the distance and confidence metrics computed on the initial latent vectors selected by different models. Values are relative to the results of the model trained without LS. Positive smoothing improves the sampling, whereas the samples selected by the negative smoothing model perform markedly worse. Fig. 6b visualizes the mean cosine similarity and their standard deviation between consecutive image gradients for the individual optimization steps. While the gradient directions are stable for models trained with hard labels and positive smoothing, the optimization path on the negative smoothing model is characterized by many changes of direction.

## 4.4 ABLATION STUDY: WHICH STAGES OF MODEL INVERSION ARE AFFECTED BY LS?

The inference process of MIAs can be grouped into three stages: latent vector sampling, optimization, and result selection. We will now individually analyze the impact of LS on each stage while isolating influences from the other stages. The analyses are based on the FaceScrub models.

**Stage 1 – Sampling (Affected):** The initial phase involves the selection of latent vectors to be optimized by the attack. PPA first samples a larger pool of latent vectors and subsequently chooses a fixed set of vectors for each target class whose corresponding images achieve the highest classification probability on the target model under random transformations. For our analysis, we created a fixed pool of 10,000 random latent vectors and then let each model select 50 samples for each class. To assess the quality of the initial sampling, we computed feature distances between the corresponding generated images and the training data, analogous to our evaluation metrics. Furthermore, we considered the confidences assigned by the evaluation model to determine which model selected samples that visually resembled the target classes most closely. The results, which are presented in Fig. 6a, again state the relative advantage compared to the model trained with hard labels. All three metrics indicate that the samples selected by the positive LS model indeed more closely resemble the target classes compared to the standard model. Conversely, the negative LS model exhibits degradation across all three metrics. To further validate the sampling's influence, we performed PPA's optimization process on the model trained without LS using the three different sets of initial latent vectors. As expected, the results confirm that samples selected with the positive LS model outperform those from the standard model, while the samples selected by the negative LS model underperform. Overall, LS seems to have a notable impact on the sampling stage of MIAs.

**Stage 2 – Optimization (Heavily Affected):** This stage comprises the attack process, wherein the latent vectors are optimized to reconstruct class-characteristic features. During this iterative procedure, the generated images $x$ are fed into the target model to compute a loss $\mathcal{L}$ based on its prediction $\mathcal{M}_{target}(x)$ for the target class $c$. The latent vectors are then updated to reduce the loss. To gain insights into the optimization stability, we sampled a set of 1,000 initial vectors and targets. At each step $t > 1$, we computed the loss gradients $\nabla \mathcal{L}^{(t)}$ w.r.t. the current images $x$ and measured their similarity to the gradients from the previous step $t - 1$ using the cosine similarity $S_C$ defined by

$$S_C\left(\nabla\mathcal{L}^{(t)}, \nabla\mathcal{L}^{(t-1)}\right) = \frac{\nabla\mathcal{L}^{(t)} \cdot \nabla\mathcal{L}^{(t-1)}}{\|\nabla\mathcal{L}^{(t)}\|\|\nabla\mathcal{L}^{(t-1)}\|} \quad \text{with} \quad \nabla\mathcal{L}^{(t)} = \nabla_{x^{(t)}} \mathcal{L}\left(\mathcal{M}_{target}(x^{(t)}), c\right) \quad (4)$$

to examine the dynamics of gradient direction changes across consecutive optimization steps. A stable optimization process implies that successive gradients should remain consistent in their direction. The visual representation of the mean similarity between consecutive gradients is depicted in Fig. 6b. Remarkably, the models trained with hard labels and positive LS exhibit a high degree of gradient similarity, indicating a stable optimization. In contrast, the negative LS model shows substantial variations in gradient directions. This connotes that the optimization frequently changes direction,

with later optimization steps pointing in orthogonal or even opposing directions. This observation is further reflected in the attack metrics, which report poor results for the optimization performed on the negative LS model and again improved results for the model trained with positive LS. Consequently, LS also substantially influences the optimization stage of MIAs with negative LS hurting its stability. The unstable gradient directions explain the attack failures on models trained with negative smoothing.

**Stage 3 – Selection (Barely Affected):** After the optimization, PPA selects a subset for each target class by filtering out those results for which the target model shows the least robust confidence. The selection is done by feeding various transformed versions of each corresponding image into the target model, computing its prediction score for the target class, and averaging across all transformations. Then, the latent vectors with the highest mean confidences are selected as attack results. To measure the effects of the various models, we took the optimization results of 200 samples per target class from the model trained without LS and trained with negative LS and then repeated the filtering approach for both sample sets on the three models to see which model selects the most promising samples. The previously observed pattern – the positive LS model improves results and the negative LS model degrades them – is still apparent, but differences are rather small. Consequently, LS seems to have only a small effect on the selection stage and all models perform rather similarly.

## 5 IMPACT, FUTURE WORK AND LIMITATIONS

Deep learning promises impressive potential in virtually all areas of our life. However, its applications have to be secure and protect user and data privacy, which can be compromised by MIAs. Our findings show that LS regularization techniques can amplify model inversion attacks – a previously unexplored dimension of privacy leakage. We show how to turn an attack leverage into an attack blocker and that LS also offers a straightforward mitigation strategy by smoothing with a negative smoothing factor, which trades model calibration and small amounts of utility for a strong defense against MIAs, particularly gradient-based attacks. Importantly, this process requires no complex adjustments to the training procedure or model architecture. An interesting direction for further research involves the investigation of other regularization methods. Another critical question is whether existing attacks can be adjusted to improve their results on negative LS models, e.g., by also taking the distance to decision boundaries during the optimization into account. Furthermore, we envision that the information reduction effects of negative LS in a model's confidence scores can also be valuable in mitigating other privacy (Shokri et al., 2017; Hintersdorf et al., 2022; Struppek et al., 2023a) or model stealing attacks (Tramèr et al., 2016), that exploit model confidences in a black-box fashion.

Nevertheless, the investigation of MIAs also faces various challenges, and our work is no exception in this regard. First, performing MIAs is very time- and resource-consuming. This limitation adds constraints on the extent to which we can conduct detailed hyperparameter analyses. We anticipate that a more exhaustive grid search of smoothing parameters and schedules could potentially yield an even more favorable balance between model utility and defense against MIAs, suggesting that our results may actually underestimate the true impact of LS. However, we found $\alpha = -0.05$ generally serves as a promising initial value and our findings present compelling evidence for the dual effects of LS within the realm of privacy. Moreover, the field of MIAs still lacks a comprehensive theoretical framework. In contrast to other attack classes, e.g., adversarial examples, MIAs are considerably more intricate and require a deeper understanding of the features learned and encoded in a model's weights. Our research is a first step in advancing the understanding of MIAs.

## 6 CONCLUSION

In contrast to previous literature on MIAs, we conducted the first study on the impact of model regularization on the attacks' success. Specifically, we investigated the impact of label smoothing regularization on the vulnerability of image classifiers to MIAs. Our findings reveal a remarkable phenomenon: training a model with a positive smoothing factor increases its privacy leakage, particularly in settings with limited training data. In contrast, training a model with negative label smoothing counteracts this trend and emerges as a practical and viable defense mechanism. No architectural modifications or complex training procedures are required, and it only slightly reduces model utility. Our work underlines the importance of delving more into factors that influence a model's privacy leakage, moving the research paradigm from improving the attacks themselves.

## ETHICS STATEMENT

This work investigates model inversion attacks, a common privacy attack against deep learning classifiers. Successful attacks bear the risk of privacy leakage and can cause serious harm to individuals in privacy-sensitive settings. Whereas we did not propose any novel attack algorithms, our insights into label smoothing regularization can indeed have practical implications for the privacy of models. However, we believe that informing the community about the dual role of label smoothing is important to raise awareness of these risks. Moreover, we also present a practical defense strategy by training classifiers with negative label smoothing to mitigate model inversion attacks and the corresponding privacy risks. Overall, we are convinced that the benefits of the present work outweigh any potential risk.

## ACKNOWLEDGMENTS

This work was supported by the German Ministry of Education and Research (BMBF) within the framework program "Research for Civil Security" of the German Federal Government, project KISTRA (reference no. 13N15343). We gratefully acknowledge support from the German Center for Artificial Intelligence (DFKI) project "SAINT".

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

# A FORMAL ANALYSIS OF LABEL SMOOTHING REGULARIZATION

We provide a more formal analysis of Label Smoothing regularization to demonstrate its impact on the model training. Be $\mathbf{y} \in \{0, 1\}^C$ with $|\mathbf{y}| = \sum_{k=1}^{C} y_k = 1$ the one-hot encoded ground-true label of a training sample $x$ with $C$ possible classes. Here, $y_k$ denotes the $k$-th entry in $\mathbf{y}$. The label vector $\mathbf{y}$ contains $C$ entries, which are all set to 0 except the correct label set to 1.

Let $\mathbf{p} \in [0, 1]^C$ with $|\mathbf{p}| = \sum_{k=1}^{C} p_k = 1$ further denote the softmax probability vector computed by the classifier for input $x$. Again, $p_k$ denotes the $k$-th entry of $\mathbf{p}$. The softmax function is computed on the output logits $\mathbf{z}$ and corresponds to the model's confidence for the $j$-th class. It is computed as

$$\sigma(\mathbf{z})_j = \frac{e^{z_j}}{\sum_{k=1}^{C} e^{z_k}} \, . \tag{5}$$

Neural network classifiers are usually trained by minimizing a cross-entropy loss, defined by

$$\mathcal{L}_{CE}(\mathbf{y}, \mathbf{p}) = -\sum_{k=1}^{C} y_k \log p_k \, . \tag{6}$$

Given the standard empirical risk minimization framework, a classifier $\mathcal{M}$ is trained by searching the hypothesis space $\mathcal{H}$ for a model $\mathcal{M}^*$ that minimizes the loss on the training set $\mathcal{S}$ with $N$ samples:

$$\mathcal{M}^* = \operatorname*{arg\,min}_{\mathcal{M} \in \mathcal{H}} \frac{1}{N} \sum_{i=1}^{N} \mathcal{L}_{CE}(\mathbf{y}^{(i)}, \mathbf{p}^{(i)}) \, . \tag{7}$$

Here, $\mathbf{y}^{(i)}$ denotes the label vector for the $i$-th training sample, and $\mathbf{p}^{(i)}$ is model's predicted confidence vector for this sample. Practically, this is done by adjusting a model's parameters with gradient descent. Training with standard cross-entropy loss encourages the model to increase the logit values for the predicted class. This can be seen by computing the gradients of $\mathcal{L}_{CE}$ with respect to the $j$-th logit value $z_j$.

First, we compute the derivative of the softmax score $p_i$ with respect to the logit value $z_j$. Let us start with the case $i = j$, i.e., the index of the softmax score and the logit vector are identical:

$$\frac{\partial p_j}{\partial z_j} = \frac{e^{z_j} \sum_{k=1}^{C} e^{z_k} - e^{z_j} e^{z_j}}{\left(\sum_{k=1}^{C} e^{z_k}\right)^2} = \frac{e^{z_j}}{\sum_{k=1}^{C} e^{z_k}} - \left(\frac{e^{z_j}}{\sum_{k=1}^{C}}\right)^2 = p_j - p_j^2 = p_j(1 - p_j) \, . \tag{8}$$

For the case $i \neq j$, the derivative is computed as follows:

$$\frac{\partial p_i}{\partial z_j} = \frac{0 - e^{z_i} e^{z_j}}{\left(\sum_{k=1}^{C} e^{z_k}\right)^2} = -p_i p_j \, . \tag{9}$$

With the softmax derivatives from Eq. (8) and Eq. (9), we can now compute the derivatives with respect to the output logits:

$$\frac{\partial \mathcal{L}_{CE}(\mathbf{y}, \mathbf{p})}{\partial p_j} = -\sum_{k=1}^{C} y_k \frac{\partial \log(p_k)}{\partial z_j} = -\sum_{k=1}^{C} \frac{y_k}{p_k} \frac{\partial p_k}{\partial z_j} = -\frac{y_j}{p_j} p_j(1 - p_j) + \sum_{k \neq j} \frac{y_k}{p_j} p_k p_j$$

$$= -y_j(1 - p_j) + p_j \sum_{k \neq j} y_k = -y_j + p_j \sum_{k=1}^{C} y_k = p_j - y_j \, . \tag{10}$$

Given the resulting gradient function for a model's logits, it becomes clear that gradient descent updates the model weights to increase the logit values for the target index $j$ while decreasing the values for all other classes.

Next, let us analyze the gradients for training with label smoothing regularization, which replaces the one-hot encoded label vector $\mathbf{y}$ with its smoothed variant $\mathbf{y}^{\text{LS}}$ defined by

$$\mathbf{y}^{\text{LS}} = (1 - \alpha) \cdot \mathbf{y} + \frac{\alpha}{C} \, . \tag{11}$$

Here, $\alpha = (-\infty, 1]$ denotes the smoothing factor. Standard label smoothing uses $\alpha > 0$, whereas negative label smoothing applies $\alpha < 0$. The case of $\alpha = 0$ corresponds to the hard label setting. Recall that label smoothing still maintains the condition $|y| = 1$, independent of the selected smoothing factor. Plugged into the logit gradient formula from Eq. (10), it allows us to analyze the impact of label smoothing during training.

First, we take a look at the gradients for the $j$-th logit vector entry corresponding to the ground-true class. In this case, label smoothing replaces $y_j = 1$ by $y_j^{\text{LS}} = 1 - \alpha + \frac{\alpha}{C}$, which leads to the following derivative:

$$\frac{\partial \mathcal{L}_{CE}(\mathbf{y}^{\text{LS}}, \mathbf{p})}{\partial p_j} = p_j - y_j^{\text{LS}} = p_j - \left(1 - \alpha + \frac{\alpha}{C}\right) . \tag{12}$$

Analogously, the gradients for the $i$-th entries in the logit vector that do not correspond to the ground-true class are computed by:

$$\frac{\partial \mathcal{L}_{CE}(\mathbf{y}^{\text{LS}}, \mathbf{p})}{\partial p_i} = p_i - y_i^{\text{LS}} = p_i - \frac{\alpha}{C} . \tag{13}$$

Given the nature of gradient descent, which subtracts the gradients (weighted by the learning rate) from each parameter, label smoothing offers interesting effects on the updates of the weights used to compute the logit vector. Generally, the weights associated with the $j$-th logit, which corresponds to the ground-true class $j$, are increased, as long as $p_j - y_j^{\text{LS}} < 0$ holds:

$$p_j - y_j^{\text{LS}} < 0 \iff \begin{cases} p_j < 1, & \text{if } \alpha = 0 \\ p_j < 1 - \alpha + \frac{\alpha}{C}, & \text{otherwise.} \end{cases} \tag{14}$$

We can see that for training without label smoothing the weights for the target class are almost always increased but the gradient update saturates when $p_j$ approaches $1.0$. For positive label smoothing, this saturation effect occurs earlier, when $p_j$ approaches $1 - \alpha + \frac{\alpha}{C}$. Let us take an example with $\alpha = 0.1$ and $C = 10$ classes. In this case, the weights are increased until $p_j = 0.91$. If the predicted confidence for a sample exceeds this value, the gradients change directions and the resulting weights are reduced. This explains the calibration effects of label smoothing, which prevents the model from being overconfident in its predictions.

For negative label smoothing, on the other hand, this saturation effect never occurs, since $1 - \alpha + \frac{\alpha}{C} > 1$ always holds true for $C > 1$ and $\alpha < 0$. Therefore, the weights for computing the logits of the target class are always increased, even if the predicted confidence approaches $1.0$. This explains why models trained with negative label smoothing are overconfident in their predictions and usually only barely calibrated.

Revert effects can be shown for the weights of the remaining logits. The weights associated with those outputs are decreased as long as the following holds true:

$$p_i - y_i^{\text{LS}} > 0 \iff \begin{cases} p_i > 0, & \text{if } \alpha = 0 \\ p_i > \frac{\alpha}{C}, & \text{otherwise.} \end{cases} \tag{15}$$

Again, training with hard labels always reduces the weights but the gradients saturate for $p_i$ approaching $0.0$. For positive label smoothing, the gradient directions change as soon as $p_i < \frac{\alpha}{C}$, supporting the mitigation of overconfidence. For negative label smoothing $p_i - y_i^{\text{LS}} > 0$ always holds true since $\frac{\alpha}{C}$ is always smaller than $0.0$ because $C > 0$ and $\alpha < 0$.

## B  EXPERIMENTAL DETAILS

Here, we state the technical details of our experiments to improve reproducibility and eliminate ambiguities.

### B.1  HARD- AND SOFTWARE DETAILS

We performed all our experiments on NVIDIA DGX machines running NVIDIA DGX Server Version 5.2.0 and Ubuntu 20.04.4 LTS. The machines have 1TB of RAM and contain NVIDIA A100-SXM4-40GB GPUs and AMD EPYC 7742 64-core CPUs. We further relied on CUDA 11.4, Python 3.8.10, and PyTorch 2.0.0 with Torchvision 0.15.1 Paszke et al. (2019) for our experiments. If not stated otherwise, we used the model architecture implementations and pre-trained ImageNet weights provided by Torchvision. We further provide a Dockerfile together with our code to make the reproduction of our results easier. In addition, all training and attack configuration files are available to reproduce the results stated in this paper. Our main experiments are built around the Plug & Play Attacks (Struppek et al., 2022a) repository available at `https://github.com/LukasStruppek/Plug-and-Play-Attacks`. Note that we updated the PyTorch version, which may lead to small differences when repeating the experiments with an older version.

### B.2  EVALUATION MODELS

For experiments based on Plug & Play Attacks (PPA), we used the pre-trained Inception-v3 evaluation models provided with the code repository at `https://github.com/LukasStruppek/Plug-and-Play-Attacks`. For training details, we refer to Struppek et al. (2022a). The models achieve a test accuracy of $96.20\%$ (FaceScrub) and $93.28\%$ (CelebA), respectively.

We also used the pre-trained FaceNet Schroff et al. (2015) from https://github.com/timesler/facenet-pytorch to measure the distance between training samples and attack results on the facial recognition tasks. The FaceNet model is based on the Inception-ResNet-v1 Szegedy et al. (2017) architecture and has been trained on VGGFace2 Cao et al. (2018).

For experiments based on CelebA classifiers with smaller image resolutions, we used the evaluation model provided at `https://github.com/SCccc21/Knowledge-Enriched-DMI` for download. The model is built around the Face.evoLVe (Wang et al., 2021b) framework with a modified ResNet50 backbone and achieves a stated test accuracy of $95.88\%$. For training details, we refer to Zhang et al. (2020).

### B.3  TARGET MODELS

For training target models with PPA, we relied on the training scripts and hyperparameters provided in the corresponding code repository and described in Struppek et al. (2022a) The only training parameter we changed was the smoothing factor of the label smoothing loss. All models were trained for 100 epochs with the Adam optimizer (Kingma & Ba, 2015) and an initial learning rate of $0.001$ and $\beta = (0.9, 0.999)$. We multiplied the learning rate after 75 and 90 epochs by a factor of $0.1$. The batch size was set to 128 during training. All data samples were normalized with $\mu = \sigma = 0.5$ and resized to $224 \times 224$. The training samples were then augmented by random cropping with a scale of $[0.85, 1.0]$ and a fixed ratio of $1.0$. Crops were then resized back to $224 \times 224$. We also applied random color jitter with brightness and contrast factors of $0.2$ and saturation and hue factors of $0.1$. Finally, samples were horizontally flipped in $50\%$ of the cases.

The target models trained on $64 \times 64$ CelebA images were trained with the training script provided at `https://github.com/SCccc21/Knowledge-Enriched-DMI`. To use a more recent and advanced architecture, we trained ResNet-50 models initialized with pre-trained ImageNet weights. The models were trained for 100 epochs with the SGD optimizer with an initial learning rate of $0.01$ and a momentum term of $0.9$. The batch size was set to 64. We reduced the learning rate after 50 and 75 epochs by a factor of $0.1$. Weight decay has not been used for training.

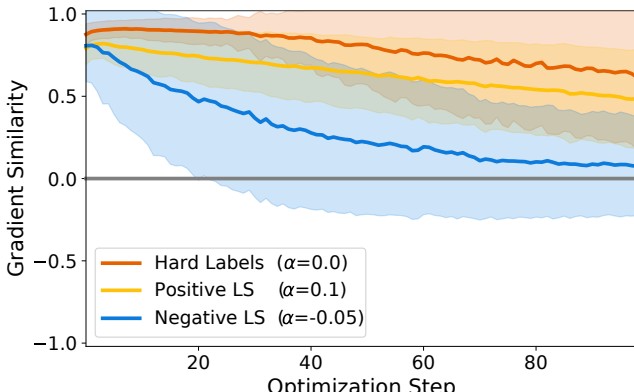

Figure 7: Similarity between consecutive image gradients based on a cross-entropy loss instead of the Poincaré loss used in Fig. 6b.

### B.4 PLUG & PLAY ATTACKS

PPA consists of three stages: latent vector sampling, optimization, and result selection. We provide a brief overview of each stage and refer for a more comprehensive introduction to Struppek et al. (2022a). The attack parameters follow those of the paper and used a pre-trained FFHQ StyleGAN 2 (Karras et al., 2020), available at `https://github.com/NVlabs/stylegan2-ada-pytorch`.

**Stage 1: Latent Vector Sampling:** PPA first samples a large number of latent vectors as candidates for the attack. Out of this set of latent vectors, the attack selects for each target class a fixed number of vectors as a starting point for the optimization. The selection is done by generating the corresponding images for each latent vector and then measuring the target model's prediction confidence on the augmented version of these images. The top $k$ samples with the highest mean confidence assigned are then selected as starting points.

During the sampling stage, we sampled 200 candidates for each target class out of a total search space of 2,000 (FaceScrub) and 5,000 (CelebA). Since the StyleGAN model generates images of size $1024 \times 1024$, samples were first center cropped with size $800 \times 800$ and then resized to $224 \times 224$.

**Stage 2: Optimization:** Instead of a standard cross-entropy loss, PPA uses a Poincaré loss function to mitigate the problem of vanishing gradients:

$$\mathcal{L}_{Poincaré} = d(u, v)$$
$$= arcosh\left(1 + \frac{2\|u - v\|_2^2}{(1 - \|u\|_2^2)(1 - \|v\|_2^2)}\right). \tag{16}$$

Here, $u = \frac{o}{\|o\|_1}$ are the normalized output logits and $v$ is the one-hot encoded target vector with the target label set to 0.9999 instead of 1.0. The attack further applies random augmentations on the images generated by the GAN before feeding them into the target model to increase the attack's robustness and avoid the generation of adversarial examples.

In our experiments, samples were optimized for 50 (FaceScrub) and 70 (CelebA) steps, respectively. Before feeding the cropped and resized samples into the target model, a random resized crop with a scale between $[0.9, 1.0]$ was applied and the cropped images resized back to $224 \times 224$. For optimizing the latent vectors, the Adam optimizer with a learning rate of 0.005 and $\beta = (0.1, 0.1)$ was used.

We repeated the experiment with a standard cross-entropy loss instead of PPA's Poincaré loss to demonstrate that the gradient instability does not arise from the model's optimization goal. We kept all other parameters identical. The results in Fig. 7 demonstrate that also with another loss function used the gradients on the model trained with negative LS are still unstable, and the similarity between consecutive gradients is still low.

**Stage 3: Result Selection:** Out of the set of optimized latent vectors for each target class, PPA selects a subset for which the target model shows the highest robustness under random augmentations.

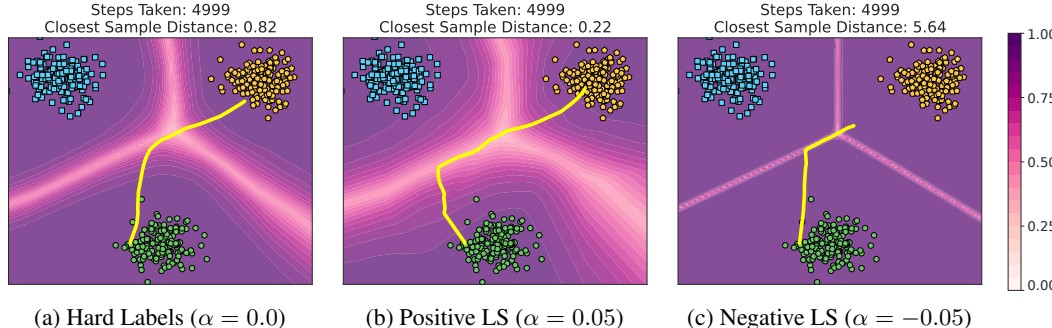

(a) Hard Labels ($\alpha = 0.0$)      (b) Positive LS ($\alpha = 0.05$)      (c) Negative LS ($\alpha = -0.05$)

Figure 8: Variation of our motivational example without using an attack-stopping criterion. Instead, the attack in all three settings is run for 5,000 steps.

More specifically, each image corresponding to an optimized latent vector undergoes strong random transformations to create variations of it. All augmented variations are then fed into the target model to compute the mean prediction confidence on the target class. Out of all candidates, the top $k$ samples with the highest robust confidence are selected as final attack results.

During the final selection stage, 50 samples out of the 200 optimized samples were selected for each target class. As random transformations horizontal flipping with $p = 0.5$ and random cropping with a scale between $[0.5, 0.9]$ and a ratio of $[0.8, 1.2]$ are performed $100$ times. The resulting samples were resized back to $224 \times 224$ before being fed into the target models.

## B.5    Comparison to Existing Defense Mechanisms

For comparison to previous defense approaches, we trained ResNet-152 models with BiDO (Peng et al., 2022) and MID (Wang et al., 2021c). The model architecture is based on the official PyTorch implementation. For MID, we added the information bottleneck based on the variational method between the average pooling and the final linear layer and set the bottleneck size to $k = 1024$. For BiDO, we used the outputs of the four major ResNet blocks as inputs for the regularization loss. We further relied on the Hilbert-Schmidt independence criterion (HSIC) as a dependency measure since the corresponding paper reported better results compared to the constrained covariance (COCO) dependency. The remaining training hyperparameters and data augmentations are identical to those stated in Appx. B.3

## B.6    Toy Example from Sec. 3

The network architecture used for the motivational example in Sec. 3 is a simple 3-layer fully-connected network. The hidden layer consists of 20 neurons. A batch norm layer is placed after the first and second layers, each followed by a ReLU activation. All models were trained with a standard cross-entropy loss (see Eq. (3)) with different label smoothing factors $\alpha \in \{0, 0.05, -0.05\}$. The models were then optimized for 5000 iterations using standard SGD with a learning rate of 0.001 and a momentum of 0.9.

The attack starts from a (fixed) random point from the *green circles* class and updates it to maximize the model's prediction score for the *orange pentagons* class by minimizing an identity loss (cross-entropy loss) as proposed by Zhang et al. (2020) (GMI). Since no image prior is applied, there are no additional loss terms. For optimization, SGD with a learning rate of $0.1$ and no momentum is used. For the experiment, we stopped the optimization as soon as the model's confidence for the class *orange pentagons* exceeds $95\%$.

In addition, we repeated the attack process without any stopping criterion and optimized the samples for 5000 steps. The results depicted in Fig. 8 draw a similar picture. For training without LS, the attack approaches the training data but the results are compared to training with positive LS markedly further away from the data. For the negative LS model, the attack again fails to approach the training set and stays close to the decision boundary.

# C    ADDITIONAL EXPERIMENTAL RESULTS

In this section, we state additional attack results that did not fit into the main part of the paper. Besides the metrics from the main paper, we also computed the common Fréchet inception distance (FID) (Heusel et al., 2017) between synthetic samples and the training data. Moreover, the knowledge extraction score $\xi_{test}$ is also computed as the surrogate model's prediction accuracy on the target model's test data. To measure a model's calibration, we computed the expected calibration error (ECE) (Naeini et al., 2015) with $10$ bins and the $l_2$ norm on the individual test splits.

## C.1    PLUG & PLAY ATTACKS RESULTS FOR VARIOUS ARCHITECTURES

Tab. 2 states the results for PPA performed against various architectures trained on FaceScrub or CelebA, respectively. We further varied the LS smoothing factor to showcase the influence of difference values. All training and attack hyperparameters are identical between the different runs. The results extend Tab. 1 and correspond to Fig. 2b from the main paper.

Table 2: Additional PPA attack results for attacks against models trained with label smoothing.

| | Architecture | $\alpha$ | ↑ Test Acc | ↓ ECE | ↑ Acc@1 | ↑ Acc@5 | ↓ $\delta_{face}$ | ↓ $\delta_{eval}$ | ↓FID | ↑ $\xi_{train}$ | ↑ $\xi_{test}$ |
|---|---|---|---|---|---|---|---|---|---|---|---|
| **FaceScrub** | **ResNet-152** | 0.3 | 96.96% | 0.3248 | 95.80% ± 2.6 | 98.47% | 0.6168 | 107.76 | 41.22 | 72.83% | 70.01% |
| | | 0.2 | 97.23% | 0.2476 | 94.00% ± 2.9 | 98.77% | 0.6567 | 109.46 | 44.10 | 71.08% | 69.67% |
| | | 0.1 | 97.39% | 0.1935 | 95.20% ± 2.9 | 98.47% | 0.6343 | 107.36 | 43.33 | 70.95% | 68.77% |
| | | 0.0 | 94.93% | 0.0619 | 94.32% ± 1.9 | 99.37% | 0.7060 | 124.30 | 40.88 | 61.19% | 58.69% |
| | | −0.005 | 94.01% | 0.1389 | 80.66% ± 2.0 | 95.82% | 0.8186 | 138.64 | 45.12 | 58.42% | 55.52% |
| | | −0.01 | 93.77% | 0.1262 | 59.87% ± 4.9 | 86.44% | 0.9225 | 158.07 | 49.84 | 53.69% | 52.03% |
| | | −0.05 | 91.45% | 0.1474 | 14.34% ± 7.6 | 30.94% | 1.2320 | 239.02 | 59.38 | 16.45% | 15.73% |
| | **ResNeXt-50** | 0.1 | 97.54% | 0.1461 | 95.95% ± 2.0 | 99.25% | 0.6346 | 107.38 | 44.57 | 73.58% | 71.30% |
| | | 0.0 | 95.25% | 0.0771 | 94.97% ± 1.6 | 99.38% | 0.6977 | 119.51 | 41.61 | 67.78% | 64.86% |
| | | −0.05 | 92.40% | 0.1048 | 9.40% ± 5.6 | 22.77% | 1.2790 | 240.69 | 66.29 | 15.31% | 15.39% |
| | **DenseNet-121** | 0.1 | 97.15% | 0.0452 | 94.85% ± 2.9 | 97.70% | 0.6416 | 110.31 | 43.18 | 67.42% | 64.81% |
| | | 0.0 | 95.72% | 0.1865 | 96.05% ± 1.2 | 99.63% | 0.6795 | 116.40 | 43.11 | 73.03% | 71.30% |
| | | −0.05 | 92.13% | 0.1525 | 40.69% ± 7.1 | 69.69% | 0.9733 | 179.53 | 49.91 | 32.79% | 31.76% |
| **CelebA** | **ResNet-152** | 0.1 | 95.11% | 0.4545 | 92.85% ± 3.4 | 96.46% | 0.6065 | 275.30 | 38.73 | 66.13% | 59.65% |
| | | 0.0 | 87.05% | 0.0899 | 81.75% ± 1.3 | 95.03% | 0.7406 | 318.09 | 36.12 | 59.77% | 50.57% |
| | | −0.05 | 83.59% | 0.2179 | 26.41% ± 3.4 | 49.96% | 1.0420 | 441.67 | 61.30 | 7.08% | 5.89% |
| | **ResNeXt-50** | 0.1 | 95.27% | 0.4002 | 93.37% ± 3.3 | 96.81% | 0.6010 | 275.48 | 38.59 | 65.75% | 58.95% |
| | | 0.0 | 87.85% | 0.1174 | 85.13% ± 1.1 | 96.07% | 0.7310 | 307.00 | 34.99 | 63.14% | 55.79% |
| | | −0.05 | 84.79% | 0.1999 | 32.83% ± 3.9 | 56.61% | 1.0140 | 430.08 | 56.62 | 33.54% | 28.86% |
| | **DenseNet-121** | 0.1 | 92.88% | 0.4113 | 90.99% ± 3.3 | 96.35% | 0.6484 | 376.46 | 89.54 | 70.84% | 62.48% |
| | | 0.0 | 86.05% | 0.0963 | 76.49% ± 1.0 | 92.23% | 0.7410 | 383.74 | 87.73 | 60.42% | 51.00% |
| | | −0.05 | 86.48% | 0.2062 | 72.70% ± 2.2 | 90.80% | 0.7866 | 467.67 | 94.78 | 50.22% | 44.07% |

## C.2 ADDITIONAL RESULTS FOR DEFENSE MECHANISMS

We compared the defensive effects of negative LS to state-to-the-art defenses MID (Wang et al., 2021c) and BiDO (Peng et al., 2022). The implementation of the defense mechanisms is based on https://github.com/AlanPeng0897/Defend_MI. We adjusted the implementation to support the ResNet-152 architecture. Training and attack hyperparameters are identical for all models. Compared to the original evaluations, we tested the defense mechanisms on high-resolution data for which both, MID and BiDO, only provide a partial defense to MIAs. Smoothing the labels with a small negative factor beats both approaches by keeping more of the model's utility while significantly decreasing the attacks' success.

Table 3: PPA results against state-the-art-defenses and negative LS as defense mechanism.

| | Defense | Defense Parameter | ↑ Test Acc | ↓ ECE | ↑ Acc@1 | ↑ Acc@5 | ↓ $\delta_{face}$ | ↓ $\delta_{eval}$ | ↓FID | ↑ $\xi_{train}$ | ↑ $\xi_{test}$ |
|---|---|---|---|---|---|---|---|---|---|---|---|
| **FaceScrub** | No Defense | $\alpha = 0.0$ | 94.93% | 0.0619 | 94.32% ± 1.9 | 99.37% | 0.7060 | 124.30 | 40.88 | 61.19% | 58.69% |
| | Label Smoothing | $\alpha = -0.05$ | 91.45% | 0.1474 | 14.34% ± 7.6 | 30.94% | 1.2320 | 239.02 | 59.38 | 16.45% | 15.73% |
| | MID | $\beta = 0.001$ | 93.14% | 0.0599 | 94.57% ± 1.6 | 99.48% | 0.6746 | 111.15 | 43.74 | 73.58% | 70.51% |
| | | $\beta = 0.005$ | 91.10% | 0.0666 | 92.04% ± 2.5 | 98.84% | 0.7163 | 115.07 | 44.75 | 72.95% | 71.17% |
| | | $\beta = 0.01$ | 90.68% | 0.0722 | 88.46% ± 3.0 | 98.01% | 0.7212 | 119.01 | 44.05 | 69.66% | 67.24% |
| | BiDO-HSIC | $\lambda = (0.05, 0.5)$ | 93.72% | 0.1042 | 89.28% ± 2.0 | 98.25% | 0.7536 | 124.20 | 44.16 | 70.58% | 67.40% |
| | | $\lambda = (0.05, 1.0)$ | 93.11% | 0.1004 | 85.60% ± 2.6 | 97.32% | 0.7796 | 130.86 | 45.09 | 67.43% | 65.47% |
| | | $\lambda = (0.05, 2.5)$ | 87.14% | 0.1506 | 45.42% ± 4.4 | 73.60% | 0.9083 | 154.49 | 49.29 | 59.30% | 55.91% |
| **CelebA** | No Defense | $\alpha = 0.0$ | 87.05% | 0.0899 | 81.75% ± 1.3 | 95.03% | 0.7406 | 318.09 | 36.12 | 59.77% | 50.57% |
| | Label Smoothing | $\alpha = -0.05$ | 83.59% | 0.2179 | 26.41% ± 3.4 | 49.96% | 1.0420 | 441.67 | 61.30 | 7.08% | 5.89% |
| | MID | $\beta = 0.001$ | 83.95% | 0.1087 | 80.21% ± 1.8 | 93.89% | 0.7460 | 292.31 | 40.02 | 75.36% | 67.01% |
| | | $\beta = 0.005$ | 80.43% | 0.1076 | 78.02% ± 2.2 | 93.38% | 0.7353 | 353.55 | 94.30 | 70.90% | 64.01% |
| | | $\beta = 0.01$ | 77.10% | 0.0907 | 73.94% ± 2.6 | 91.52% | 0.7352 | 351.79 | 97.44 | 70.76% | 64.18% |
| | BiDO-HSIC | $\lambda = (0.05, 0.5)$ | 79.89% | 0.2034 | 63.68% ± 1.5 | 85.92% | 0.8135 | 382.40 | 90.71 | 60.64% | 52.26% |
| | | $\lambda = (0.05, 1.0)$ | 75.83% | 0.2190 | 43.34% ± 1.3 | 69.10% | 0.8822 | 387.22 | 91.58 | 56.43% | 46.67% |
| | | $\lambda = (0.05, 2.5)$ | 58.62% | 0.2917 | 8.13% ± 0.9 | 21.83% | 1.1340 | 423.09 | 46.29 | 41.35% | 30.93% |

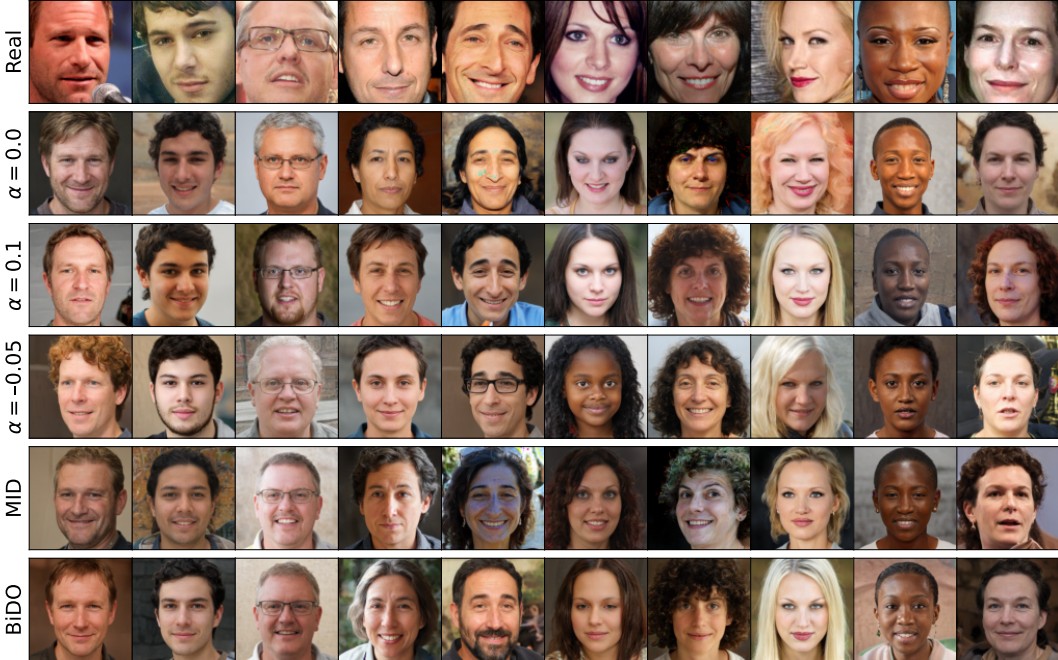

Figure 9: Attack samples from ResNet-152 models trained on FaceScrub. Samples are not cherry-picked but show the most robust attack results based on PPA's selection procedure. Results for MID and BiDO are taken from the models trained with $\lambda = 0.005$ and $\lambda = (0.05, 0.5)$, respectively.

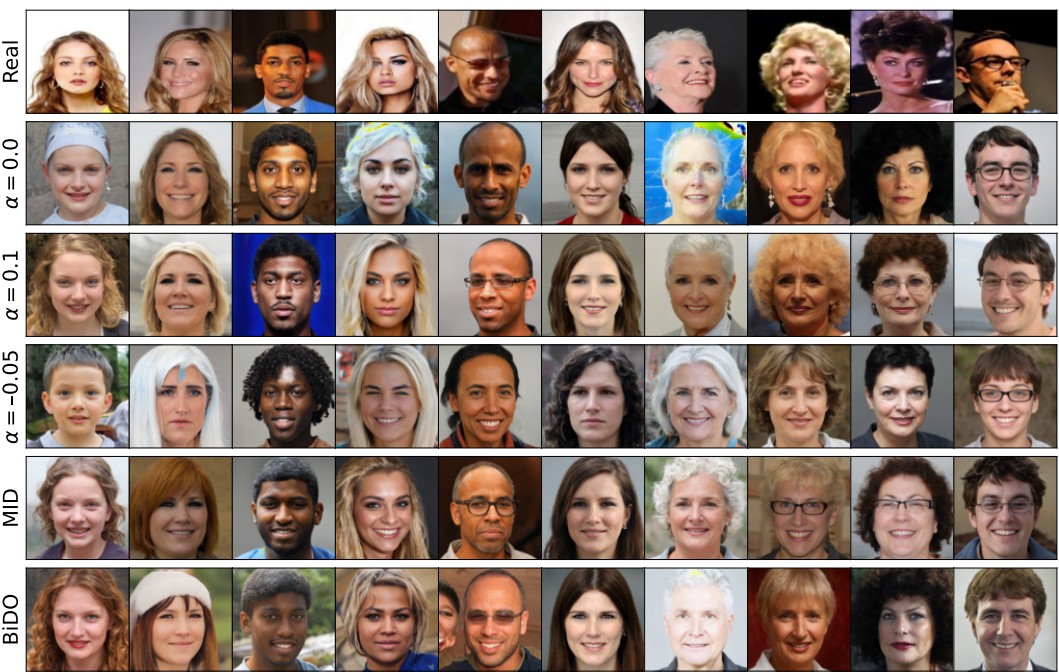

Figure 10: Attack samples from ResNet-152 models trained on CelebA. Samples are not cherry-picked but show the most robust attack results based on PPA's selection procedure. Results for MID and BiDO are taken from the models trained with $\lambda = 0.005$ and $\lambda = (0.05, 0.5)$, respectively.

## C.3 Varying Number of Training Samples

We compared the impact of training target models with LS for a varying number of training samples available. More specifically, we sampled a fixed number of samples for each class of the training data and trained the models on these smaller datasets. Whereas the impact of positive LS on a model's privacy leakage is larger for settings with fewer training samples available, negative LS improves the defense in settings with more data. Tab. 4 states the results for attacks against the various models. The results correspond to Fig. 2a from the main paper.

Table 4: PPA results against models trained on a varying number of samples per class.

| | # Samples | $\alpha$ | ↑ Test Acc | ↓ ECE | ↑ Acc@1 | ↑ Acc@5 | ↓ $\delta_{face}$ | ↓ $\delta_{eval}$ | ↓ FID | ↑ $\xi_{train}$ | ↑ $\xi_{test}$ |
|---|---|---|---|---|---|---|---|---|---|---|---|
| **FaceScrub** | **10** | 0.1 | 81.52% (+15.44) | 0.5367 | 60.38% (+17.81) | 83.81% | 0.8532 (-0.08) | 145.06 | 41.99 | 45.78% | 42.37% |
| | | 0.0 | 66.08% | 0.2108 | 42.57% | 71.34% | 0.9357 | 158.80 | 39.88 | 37.64% | 34.93% |
| | | −0.05 | 61.30% (-4.78) | 0.4265 | 32.91% (-9.66) | 60.58% | 0.9890 (+0.05) | 166.17 | 43.30 | 37.89% | 36.22% |
| | **20** | 0.1 | 93.16% (+9.03) | 0.4392 | 87.26% (+14.43) | 95.59% | 0.7155 (-0.11) | 126.03 | 42.50 | 59.92% | 56.89% |
| | | 0.0 | 84.13% | 0.1079 | 72.83% | 92.24% | 0.8291 | 142.74 | 42.85 | 55.55% | 52.03% |
| | | −0.05 | 82.47% (-1.66) | 0.2483 | 40.11% (-32.72) | 69.11% | 0.9818 (+0.15) | 170.93 | 50.59 | 48.09% | 45.78% |
| | **30** | 0.1 | 95.91% (+7.13) | 0.3773 | 92.60% (+8.5) | 97.41% | 0.6636 (-0.11) | 116.99 | 42.76 | 65.69% | 62.70% |
| | | 0.0 | 88.78% | 0.1028 | 84.10% | 96.87% | 0.7748 | 135.00 | 43.55 | 58.92% | 56.02% |
| | | −0.05 | 87.67% (-1.11) | 0.1850 | 24.53% (-59.57) | 51.15% | 1.0800 (+0.31) | 188.27 | 55.59 | 33.84% | 32.02% |
| | **40** | 0.1 | 97.02% (+4.78) | 0.3247 | 94.40% (+4.27) | 97.67% | 0.6314 (-0.11) | 111.84 | 43.22 | 67.72% | 65.42% |
| | | 0.0 | 92.24% | 0.0750 | 90.13% | 98.64% | 0.7429 | 131.69 | 41.79 | 59.96% | 56.63% |
| | | −0.05 | 89.02% (-3.22) | 0.1622 | 12.87% (-77.26) | 31.35% | 1.1950 (+0.45) | 217.94 | 59.89 | 22.38% | 21.96% |
| | **50** | 0.1 | 96.70% (+4.25) | 0.3207 | 94.54% (+5.31) | 98.14% | 0.6399 (-0.12) | 114.65 | 41.84 | 67.23% | 64.73% |
| | | 0.0 | 92.45% | 0.0680 | 89.23% | 98.15% | 0.7593 | 131.66 | 41.75 | 59.19% | 56.36% |
| | | −0.05 | 88.28% (-4.17) | 0.1808 | 18.94% (-70.29) | 42.81% | 1.1340 (+0.37) | 190.76 | 58.99 | 32.29% | 31.02% |
| **CelebA** | **10** | 0.1 | 78.96% (+20.07) | 0.5768 | 60.40% (+24.06) | 80.61% | 0.8335 (-0.10) | 343.67 | 38.46 | 47.65% | 38.88% |
| | | 0.0 | 58.89% | 0.2346 | 36.34% | 63.42% | 0.9348 | 377.917 | 33.705 | 39.90% | 32.19% |
| | | −0.05 | 57.06% (-1.83) | 0.4630 | 33.13% (-3.21) | 58.75% | 0.9740 (+0.04) | 396.79 | 38.15 | 38.12% | 31.32% |
| | **20** | 0.1 | 93.24% (+11.02) | 0.5141 | 88.79% (+17.17) | 95.06% | 0.6592 (-0.13) | 289.63 | 38.14 | 62.21% | 55.19% |
| | | 0.0 | 82.22% | 0.1126 | 71.62% | 90.29% | 0.7912 | 333.47 | 34.34 | 56.06% | 47.57% |
| | | −0.05 | 80.36% (-1.86) | 0.2600 | 35.57% (-36.05) | 61.11% | 0.9888 (+0.20) | 416.94 | 54.02 | 34.16% | 30.89% |
| | **30** | 0.1 | 92.21% (+11.32) | 0.5180 | 91.61% (+14.92) | 96.68% | 0.6166 (-0.16) | 269.37 | 37.46 | 60.52% | 54.39% |
| | | 0.0 | 80.89% | 0.1393 | 76.69% | 92.47% | 0.7757 | 326.20 | 34.01 | 57.26% | 46.80% |
| | | −0.05 | 78.23% (-2.66) | 0.2458 | 17.69% (-59.00) | 36.76% | 1.1200 (+0.34) | 472.90 | 64.22 | 22.83% | 20.71% |

## C.4 ANALYSIS OF ATTACK STAGES 4.4

Here, we provide additional numerical results for our attack stage analysis in Sec. 4.4. More specifically, we investigated three ResNet-152 models trained on FaceScrub with hard labels, positive or negative LS. Tab. 5 states the results for PPA performed on the FaceScrub ResNet-152 model trained without LS but with initial latent vectors selected by different models. Tab. 6 further states attack results for which a fixed set of random latent vectors has been optimized with the different target models. Finally, Tab. 7 contains results for which the three models selected a subset of the attack results computed on the model trained with hard labels or negative LS.

Table 5: Evaluation metrics for PPA performed with the same ResNet-152, which has been trained on FaceScrub without label smoothing. The only difference between the runs is the initial latent vectors used during the attack optimization. For each target class, 50 latent vectors have been selected from a total of 10,000 candidates. The samples were selected by three different ResNet-152, once trained without label smoothing, once with positive smoothing ($\alpha = 0.1$), and once with negative smoothing ($\alpha = -0.05$). No final selection of the results was performed.

| Smoothing $\alpha$ of Sampling Model | ↑ Acc@1 | ↑ Acc@5 | ↓ $\delta_{face}$ | ↓ $\delta_{eval}$ | ↓FID | ↑ $\xi_{train}$ | ↑ $\xi_{test}$ |
|---|---|---|---|---|---|---|---|
| 0.1 | 86.09% (+2.75) | 96.58% | 0.7764 (-0.03) | 127.92 | 42.54 | 79.98% | 75.79% |
| 0.0 | 83.34% | 95.42% | 0.8057 | 127.51 | 41.99 | 78.55% | 73.92% |
| −0.05 | 70.20% (-13.14) | 85.48% | 0.8808 (+0.08) | 142.23 | 44.47 | 77.70% | 72.89% |

Table 6: Evaluation metrics for PPA performed with three different ResNet-152, trained on FaceScrub with and without label smoothing. For each target class, 50 latent vectors have been selected from a total of 10,000 candidates by the ResNet-152 trained with hard labels. The runs differ only in the target model used to optimize the latent vectors. No final selection of the results was performed.

| Smoothing $\alpha$ of Optimization Model | ↑ Acc@1 | ↑ Acc@5 | ↓ $\delta_{face}$ | ↓ $\delta_{eval}$ | ↓FID | ↑ $\xi_{train}$ | ↑ $\xi_{test}$ |
|---|---|---|---|---|---|---|---|
| 0.1 | 85.37% (+2.03) | 93.71% | 0.7476 (-0.06) | 114.12 | 41.42 | 83.48% | 79.38% |
| 0.0 | 83.34% | 95.42% | 0.8057 | 127.51 | 41.99 | 78.55% | 73.92% |
| −0.05 | 17.14% (-66.20) | 38.76% | 1.1640 (+0.34) | 205.77 | 48.82 | 36.63% | 35.45% |

Table 7: Evaluation metrics for PPA performed on ResNet-152 models trained on FaceScrub with hard labels or negative LS, respectively. Each attack produces 200 optimized vectors for each target class. The attack's final result selection stage was then performed on models trained with different smoothing factors. All metrics were then computed on those selected subsets, each consisting of 50 samples per class.

| Optimization Model | Smoothing $\alpha$ of Selection Model | ↑ Acc@1 | ↑ Acc@5 | ↓ $\delta_{face}$ | ↓ $\delta_{eval}$ | ↓FID | ↑ $\xi_{train}$ | ↑ $\xi_{test}$ |
|---|---|---|---|---|---|---|---|---|
| | 0.1 | 96.55% (+1.98) | 99.73% | 0.6638 (-0.04) | 118.03 | 40.10 | 68.88% | 66.61% |
| $\alpha = 0.0$ | 0.0 | 94.57% | 99.39% | 0.7051 | 124.50 | 40.77 | 67.74% | 64.60% |
| | −0.05 | 94.92% (+0.35) | 99.49% | 0.6860 (-0.02) | 120.81 | 40.79 | 67.74% | 64.60% |
| | 0.1 | 17.98% (+0.23) | 41.47% | 1.1210 (-0.01) | 213.16 | 54.33 | 19.56% | 19.24% |
| $\alpha = -0.05$ | 0.0 | 17.75% | 40.42% | 1.1290 | 210.11 | 53.19 | 21.92% | 20.93% |
| | −0.05 | 14.36% (-3.39) | 30.93% | 1.2320 (-0.10) | 239.36 | 59.38 | 15.58% | 14.65% |

## C.5 Results on Low-Resolution Model Inversion Attacks

In addition to conducting experiments on high-resolution data, we also explored the impact of LS on low-resolution MIAs. More specifically, we conducted Generative MIA (GMI) (Zhang et al., 2020), Knowledge Enriched MIA (KED) (Chen et al., 2021), Logit MAximization & Model Augmentation (LOMMA) (Nguyen et al., 2023), Pseudo Label-Guided MIA (PLG-MI)(Yuan et al., 2023), Reinforcement Learning-Based MIA (RLB-MI) (Han et al., 2023), and Boundary Repulsion MIA (BREP-MI) (Kahla et al., 2022). Our experiments followed the standard evaluation protocol of the papers using $64 \times 64$ CelebA images from the 1,000 identities with the highest sample count and training VGG-16 target models. Both the training and attack phases were done using the official attack implementations. We only adjusted the target model training by adding the smoothing factors and setting the number of training epochs to 100. Additionally, for training with negative LS, we applied the same smoothing scheduler as employed in the high-resolution experiments. All attack hyperparameters remained at their default settings. For GMI, KED, LOMMA, and BREP-MI, we carried out attacks on all 1,000 identities. For PLG-MI, we followed the paper and attacked 300 identities. However, for RLB-MI, due to the extensive time requirements of the attack, we adopted the evaluation procedure outlined in the original paper and targeted only 100 randomly selected identities.

We compare the effect of LS to training with the MID (Wang et al., 2021c) and BiDO (HSIC) (Peng et al., 2022) defenses. Following the original papers, we trained two models for each defense. For MID, we used $\beta = 0.003$ and $\beta = 0.01$ as defense parameters, and for BiDO $\lambda = (0.05, 0.5)$ and $\lambda = (0.05, 1.0)$, respectively.

Tab. 8 states the evaluation results of the different attacks. Here, $\delta_{KNN}$ denotes the k-nearest neighbor distance, which states the shortest distance from the attack results to the training data from the target class. The distance is measured as the $\ell_2$ distance in the evaluation model's feature space. As evaluation model acts as a pre-trained FaceNet model. We refer to Zhang et al. (2020) for more details on this metric. We emphasize that RLB-MI and BREP-MI only compute the attack accuracy, which is why the FID score and $\delta_{KNN}$ are not stated for these attacks. Both attacks also provide no standard deviations.

Our findings consistently revealed that for most of the attacks, the positive LS amplifies privacy leakage, while negative LS mitigates this effect. In the case of the label-only BREP-MI, LS appeared to have a negligible impact on the attack results. We hypothesize that this phenomenon is attributable to the optimization strategy employed in BREP-MI, which solely relies on distance estimations to decision boundaries. LS, in contrast, influences the information content within the target model's logits and prediction scores. This explains why MIAs that rely on these components for optimization are more noticeably influenced by the choice of smoothing procedure. The introduction of boundary-based guidance into the optimization strategies of gradient-based and black-box attacks represents an intriguing avenue for further research, holding the potential to enhance the effectiveness of existing attacks. Also, negative LS has no noticeable impact on the success of PLG-MI, where the attack results are comparable to the model trained with hard labels. However, when qualitatively analyzing the attack samples in Fig. 11, it becomes clear that the attack against the negative LS model mainly produces low-quality samples without much variance between the samples, i.e., the attack, in some sense, collapsed into an adversarial sample.

Comparing the defensive effects of negative LS with MID and BiDO, we can conclude that negative LS provides better defense against most attack algorithms while keeping a similar model utility. For stronger MID and BiDO defense parameters, substantially more model utility is lost to enable a better defense. Overall, we can conclude that negative LS at least offers the same utility-privacy as MID and BiDO with the tendency to outperform it in the low-resolution setting.

Table 8: Results of MIAs against VGG-16 models trained on CelebA with $64 \times 64$ resolution.

| Attack | Type | Training | Parameter | ↑ Test Acc | ↑ Acc@1 | ↑ Acc@5 | ↓ $\delta_{KNN}$ | ↓ FID |
|---|---|---|---|---|---|---|---|---|
| **GMI**
(Zhang et al., 2020) | White-Box | Baseline
LS
LS
MID
MID
HSIC
HSIC | $\alpha = 0.0$
$\alpha = 0.1$
$\alpha = -0.05$
$\beta = 0.003$
$\beta = 0.01$
$\lambda = (0.05, 0.5)$
$\lambda = (0.05, 1.0)$ | 85.74%
88.10% (+2.36)
80.02% (-5.72)
77.56% (-8.18)
67.45% (-18.29)
79.06% (-6.68)
70.18% (-15.56) | 16.00% ± 3.75
25.40% ± 5.06 (+9.40)
5.92% ± 2.31 (-10.08)
14.60% ± 2.80 (-1.40)
12.56% ± 3.79 (-3.44)
7.40% ± 2.79 (-8.60)
2.76% ± 1.36 (-13.24) | 36.60% ± 4.37
47.40% ± 5.09
19.80% ± 3.91
30.00% ± 4.15
31.00% ± 3.69
17.80% ± 4.55
8.60% ± 2.21 | 1043.22
1067.55
1078.57
1079.17
1085.05
1142.14
1227.64 | 52.90
51.12
70.87
56.52
59.41
65.36
81.60 |
| **KED**
(Chen et al., 2021) | White-Box | Baseline
LS
LS
MID
MID
HSIC
HSIC | $\alpha = 0.0$
$\alpha = 0.1$
$\alpha = -0.05$
$\beta = 0.003$
$\beta = 0.01$
$\lambda = (0.05, 0.5)$
$\lambda = (0.05, 1.0)$ | 85.74%
88.10% (+2.36)
80.02% (-5.72)
77.56% (-8.18)
67.45% (-18.29)
79.06% (-6.68)
70.18% (-15.56) | 43.64% ± 3.67
68.88% ± 3.23 (+25.24)
24.10% ± 3.06 (-19.54)
60.40% ± 2.49 (+16.76)
50.44% ± 2.21 (+6.80)
42.72% ± 4.25 (-0.92)
29.00% ± 5.13 (-14.64) | 71.80% ± 3.41
86.20% ± 2.43
54.80% ± 2.88
87.80% ± 2.32
79.20% ± 1.80
71.60% ± 2.67
58.20% ± 2.60 | 897.54
791.11
953.74
797.99
821.01
865.94
932.78 | 42.59
24.10
43.56
27.58
27.02
29.98
31.95 |
| **LOMMA (GMI)**
(Nguyen et al., 2023) | White-Box | Baseline
LS
LS
MID
MID
HSIC
HSIC | $\alpha = 0.0$
$\alpha = 0.1$
$\alpha = -0.05$
$\beta = 0.003$
$\beta = 0.01$
$\lambda = (0.05, 0.5)$
$\lambda = (0.05, 1.0)$ | 85.74%
88.10% (+2.36)
80.02% (-5.72)
77.56% (-8.18)
67.45% (-18.29)
79.06% (-6.68)
70.18% (-15.56) | 53.64% ± 4.64
50.96% ± 3.52 (-2.68)
39.16% ± 4.25 (-14.48)
32.92% ± 3.59 (-20.72)
17.16% ± 3.71 (-36.48)
47.84% ± 4.32 (-5.80)
30.84% ± 4.27 (-22.80) | 79.60% ± 3.55
71.80% ± 4.48
68.00% ± 4.49
57.40% ± 3.16
36.80% ± 6.47
74.40% ± 4.51
54.20% ± 4.68 | 878.36
955.44
854.10
961.51
1044.63
892.00
963.49 | 42.28
47.44
39.24
51.98
59.07
41.76
44.36 |
| **LOMMA (KED)**
(Nguyen et al., 2023) | White-Box | Baseline
LS
LS
MID
MID
HSIC
HSIC | $\alpha = 0.0$
$\alpha = 0.1$
$\alpha = -0.05$
$\beta = 0.003$
$\beta = 0.01$
$\lambda = (0.05, 0.5)$
$\lambda = (0.05, 1.0)$ | 85.74%
88.10% (+2.36)
80.02% (-5.72)
77.56% (-8.18)
67.45% (-18.29)
79.06% (-6.68)
70.18% (-15.56) | 72.96% ± 1.29
76.52% ± 1.31 (+3.56)
63.60% ± 1.37 (-9.36)
63.56% ± 1.11 (-9.40)
50.68% ± 1.49 (-22.28)
65.68% ± 1.41 (-7.28)
47.64% ± 1.28 (-25.32) | 93.00% ± 0.83
92.40% ± 1.05
86.60% ± 0.70
90.60% ± 0.68
76.60% ± 1.58
86.60% ± 0.67
73.80% ± 0.94 | 791.80
780.76
784.43
792.74
831.31
810.61
860.08 | 33.39
33.01
40.97
39.69
36.87
35.95
38.77 |
| **PLG-MI**
(Yuan et al., 2023) | White-Box | Baseline
LS
LS
MID
MID
HSIC
HSIC | $\alpha = 0.0$
$\alpha = 0.1$
$\alpha = -0.05$
$\beta = 0.003$
$\beta = 0.01$
$\lambda = (0.05, 0.5)$
$\lambda = (0.05, 1.0)$ | 85.74%
88.10% (+2.36)
80.02% (-5.72)
77.56% (-8.18)
67.45% (-18.29)
79.06% (-6.68)
70.18% (-15.56) | 71.00% ± 3.31
80.00% ± 4.47 (+9.00)
72.00% ± 2.50 (+1.00)
72.00% ± 6.08 (+1.00)
59.00% ± 2.45 (-12.00)
70.00% ± 4.00 (-1.00)
42.00% ± 4.80 (-29.00) | 92.00% ± 3.16
92.00% ± 3.16
89.00% ± 2.00
89.00% ± 3.00
81.00% ± 3.87
85.00% ± 3.16
62.00% ± 5.10 | 1358.56
1329.05
1544.82
1378.50
1487.30
1433.48
1585.35 | 22.43
21.89
78.98
20.74
20.64
25.37
30.52 |
| **RLB-MI**
(Han et al., 2023) | Black-Box | Baseline
LS
LS
MID
MID
HSIC
HSIC | $\alpha = 0.0$
$\alpha = 0.1$
$\alpha = -0.05$
$\beta = 0.003$
$\beta = 0.01$
$\lambda = (0.05, 0.5)$
$\lambda = (0.05, 1.0)$ | 85.74%
88.10% (+2.36)
80.02% (-5.72)
77.56% (-8.18)
67.45% (-18.29)
79.06% (-6.68)
70.18% (-15.56) | 52.00%
65.00% (+13.00)
19.00% (-33.00)
27.00% (-25.00)
20.00% (-32.00)
35.00% (-17.00)
25.00% (-27.00) | 75.00%
84.00%
48.00%
41.00%
42.00%
57.00%
49.00% | -
-
-
-
-
-
- | -
-
-
-
-
-
- |
| **BREP-MI**
(Kahla et al., 2022) | Label-Only | Baseline
LS
LS
MID
MID
HSIC
HSIC | $\alpha = 0.0$
$\alpha = 0.1$
$\alpha = -0.05$
$\beta = 0.003$
$\beta = 0.01$
$\lambda = (0.05, 0.5)$
$\lambda = (0.05, 1.0)$ | 85.74%
88.10% (+2.36)
80.02% (-5.72)
77.56% (-8.18)
67.45% (-18.29)
79.06% (-6.68)
70.18% (-15.56) | 49.00%
56.33% (+7.33)
48.70% (-0.30)
53.43% (+4.43)
43.17% (-5.83)
42.60% (-6.40)
27.10% (-21.9) | 73.67%
77.00%
70.50%
76.05%
69.25%
66.90%
50.30% | -
-
-
-
-
-
- | -
-
-
-
-
-
- |

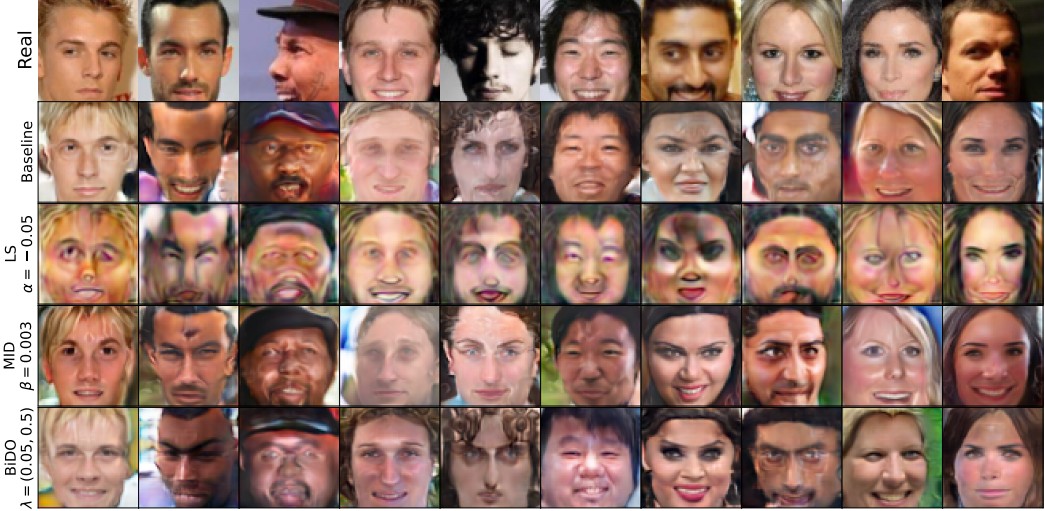

Figure 11: Visualization of attack results for PLG-MI performed against models trained with hard labels (Baseline), negative LS, MID, and BiDO as defenses. Whereas the attack accuracy for the models trained with hard labels and negative LS indicate comparable attack success, the attack results for the negative LS models have substantially reduced image quality and, consequently, reveal fewer private features.

## C.6 Investigating the Impact of Prediction Accuracy

Classifiers trained with positive LS often improve their prediction accuracy on unseen test data compared to models trained with standard cross-entropy loss. Analogously, negative LS training can reduce a model's prediction accuracy. To investigate if the defensive effect of negative LS on MIAs arises from a reduced model performance, we re-trained ResNet-101 models targeting the same test accuracy as the negative LS models achieved. More specifically, we trained models without LS ($\alpha = 0.0$) and with positive LS ($\alpha = 0.1$) and stopped the training as soon as the models reached the same test accuracy as the model trained with negative LS ($\alpha = -0.05$). Since we measured the accuracy after each training epoch, the resulting test accuracy is slightly higher for the models trained with $\alpha \geq 0.0$ but still in the same range as the negative LS models. Importantly, the models trained with positive LS already achieved the target test accuracy after only 4 (FaceScrub) and 7 (CelebA) training epochs, respectively. In particular, the FaceScrub model has seen substantially fewer samples than the other models. It explains the attack results being slightly below those of the model trained without any LS applied, which was trained for 19 epochs.

Still, the results clearly indicate that training with negative LS leads to markedly lower attack metrics in all cases, even if the models' test accuracy is comparable. We, therefore, conclude that a model's slightly reduced prediction accuracy is not the main reason for the defensive effect of negative LS. Still, it is also clear that models with poor generalization and understanding of class characteristics can only leak less information about individual classes. Consequently, when attacking the models trained with early stopping, the attack metrics degrade compared to the models trained for all epochs. However, the gap between those models and the negative LS models is still substantially large, indicating the strong defense effect of negative LS.

Currently, results are reported for 50 gender-balanced classes in each setting. The results for targeting all classes will be added as soon as they become available. Since the *Knowledge Extraction Score $\xi$* is computed over all classes in a dataset, we will add those values later to avoid misleading results.

Table 9: PPA results against models trained with early stopping to achieve similar prediction accuracy results on the test data. Even for matching test accuracy, the attacks achieve much higher success on models trained with standard cross-entropy loss or with positive LS.

| | Defense | $\alpha$ | ↑ Test Acc | ↑ Acc@1 | ↑ Acc@5 | ↓ $\delta_{face}$ | ↓ $\delta_{eval}$ | ↓ FID | ↑ $\xi_{train}$ | ↑ $\xi_{test}$ |
|---|---|---|---|---|---|---|---|---|---|---|
| **FaceScrub** | **No Label Smoothing** | 0.0 | 91.74% | 89.68% | 99.00% | 0.7382 | 133.12 | 50.43 | 24.60% | 23.68% |
| | **Positive Label Smoothing** | 0.1 | 92.13% | 87.16% | 97.20% | 0.7766 | 135.73 | 49.56 | 25.40% | 27.19% |
| | **Negative Label Smoothing** | −0.05 | 91.45% | 14.34% | 30.94% | 1.2320 | 239.02 | 59.38 | 16.45% | 15.73% |
| **CelebA** | **No Label Smoothing** | 0.0 | 84.02% | 69.36% | 90.60% | 0.7899 | 344.82 | 49.60 | 22.90% | 20.89% |
| | **Positive Label Smoothing** | 0.1 | 86.78% | 76.64% | 92.84% | 0.785 | 332.53 | 50.05 | 20.43% | 18.35% |
| | **Negative Label Smoothing** | −0.05 | 83.59% | 26.41% | 49.96% | 1.0420 | 441.67 | 61.30 | 7.08% | 5.89% |

## C.7 CHALLENGING THE DEFENSIVE EFFECT OF NEGATIVE LABEL SMOOTHING

We further investigated if the defensive effect of negative LS on MIAs, particularly PPA in the high-resolution setting, is due to the design of the attack algorithm and if attack adjustments are able to break the defense. For this, we investigated the following settings:

- We increased the number of initially sampled latent vectors to $10,000$.

- We repeated the attack with a lower learning rate of $0.0005$.

- We repeated the attack with a higher learning rate of $0.05$.

- We replaced the Poincaré loss with a standard cross-entropy loss, following (Zhang et al., 2020).

- We added a prior loss based on the GAN's discriminator weighted by $0.01$, following Zhang et al. (2020).

- We replaced the Poincaré loss with a logit identity loss including a regularization term $p_{reg} \sim \mathcal{N}(\mu_{pen}, \sigma_{pen})$, following the implementation of Nguyen et al. (2023).

In all settings, we only changed a single aspect of the attack and kept all other parameters fixed. All experiments were conducted on the same ResNet-101 models trained with negative LS ($\alpha = -0.05$).

Table 10: Attack results for various modification of PPA's optimization method.

| | Variant | ↑Acc@1 | ↑Acc@5 | ↓$\delta_{face}$ | ↓$\delta_{eval}$ | ↓FID | ↑$\xi_{train}$ | ↑$\xi_{test}$ |
|---|---|---|---|---|---|---|---|---|
| | **Default** | 14.34% | 30.94% | 1.2320 | 239.02 | 59.38 | 16.45% | 15.73% |
| | **Increased Latent Vector Sampling** | 19.24% | 39.64% | 1.178 | 225.39 | 63.87 | 16.92% | 16.67% |
| FaceScrub | **Lower Learning Rate** | 2.72% | 9.40% | 1.381 | 222.13 | 75.94 | 10.17% | 8.77% |
| | **Higher Learning Rate** | 19.44% | 36.60% | 1.258 | 1031.79 | 86.44 | 17.11% | 15.79% |
| | **Cross-Entropy Loss** (Zhang et al., 2020) | 23.56% | 55.92% | 1.01 | 156.79 | 62.39 | 25.72% | 25.15% |
| | **Prior Loss** (Zhang et al., 2020) | 12.04% | 29.60% | 1.224 | 218.58 | 64.83 | 15.33% | 12.57% |
| | **Identity Loss** (Nguyen et al., 2023) | 22.84% | 40.56% | 1.197 | 186.52 | 84.48 | 13.89% | 12.87% |

## C.8 Impact of Label Smoothing Training on Adversarial Robustness

In this section, we investigate if training with LS has an impact on a model's robustness to adversarial examples. Adversarial examples are slightly perturbed inputs that seem benign to the human eye but can change a model's prediction substantially. In recent years, various attack algorithms and settings have been proposed (Szegedy et al., 2014; Goodfellow et al., 2015; Struppek et al., 2022b). For evaluation, we apply the following attack algorithms for crafting adversarial examples:

- **Fast Gradient Sign Method (FGSM)** (Goodfellow et al., 2015): Single-step, white-box attack. Hyperparameters: $\epsilon = 8/255$.
- **Projected Gradient Descent (PGD)** (Madry et al., 2018): Multi-step, white-box attack. Hyperparameters: $\epsilon = 8/255$, step size = $2/255$, steps = $10$, random start = True.
- **Basic Iterative Method (BIM)** (Kurakin et al., 2017): Multi-step, white-box attack. Hyperparameters: $\epsilon = 8/255$, step size = $2/255$, steps = $10$.
- **One-Pixel-Attack** (Su et al., 2019): Multi-step, black-box attack. Hyperparameters: pixels = $1$, steps = $10$, population size = $10$.

All attacks were performed on the unseen test data. We repeated each attack in a targeted and untargeted setting. In the untargeted case, an attack is successful if the prediction of the true label is prevented under adversarial perturbations. For the more challenging targeted case, we randomly selected a target label for each input. An attack is successful if the added perturbations are able to force the model to predict the target label. For both settings, we compute the attacks' success rate, i.e., the lower the success rate, the more robust a model is to adversarial perturbations.

The results in Tab. 11 demonstrate that training a model with positive LS can make a model more robust to adversarial examples. For instance, the success rate of FGSM decreases on almost all models compared to training without any LS. However, training with negative LS can have an even higher impact on a model's robustness to adversarial perturbations. The attack success rates for untargeted attacks are substantially lower in almost every case compared to models trained without LS or positive LS. Training a model with negative LS, therefore, not only makes MIAs harder to perform but also makes models more robust to adversarial examples.

Table 11: Adversarial robustness of models against various targeted and untargeted adversarial attacks. We computed the attack success rates in all cases. The lower the success rate, the more robust a model is to the individual attack algorithm.

| | Architecture | $\alpha$ | ↑ Clean Test Acc | Untargeted Attacks | | | | Targeted Attacks | | | |
|---|---|---|---|---|---|---|---|---|---|---|---|
| | | | | ↓ FGSM | ↓ PGM | ↓ BIM | ↓ One-Pixel | ↓ FGSM | ↓ PGM | ↓ BIM | ↓ One-Pixel |
| **FaceScrub** | **ResNet-152** | 0.1 | 97.39% | 70.09% | 100.00% | 100.00% | 3.93% | 7.76% | 92.42% | 93.51% | 0.00% |
| | | 0.0 | 94.93% | 96.09% | 100.00% | 100.00% | 7.63% | 55.62% | 99.95% | 99.97% | 0.03% |
| | | −0.05 | 91.45% | 13.38% | 13.94% | 13.44% | 10.43% | 22.60% | 98.52% | 98.89% | 0.08% |
| | **ResNeXt-50** | 0.1 | 97.54% | 59.21% | 100.00% | 100.00% | 3.54% | 6.84% | 83.58% | 85.56% | 0.03% |
| | | 0.0 | 95.25% | 97.36% | 100.00% | 100.00% | 7.79% | 52.46% | 100.00% | 99.97% | 0.05% |
| | | −0.05 | 92.40% | 11.17% | 11.38% | 11.11% | 9.74% | 24.00% | 99.55% | 99.74% | 0.08% |
| | **DenseNet-121** | 0.1 | 97.15% | 81.76% | 100.00% | 100.00% | 3.83% | 9.16% | 93.11% | 94.59% | 0.03% |
| | | 0.0 | 95.72% | 97.02% | 100.00% | 100.00% | 5.44% | 39.73% | 100.00% | 100.00% | 0.05% |
| | | −0.05 | 92.13% | 31.84% | 32.74% | 32.15% | 9.61% | 18.72% | 86.17% | 88.52% | 0.08% |
| **CelebA** | **ResNet-152** | 0.1 | 95.11% | 97.44% | 100.00% | 100.00% | 6.49% | 17.64% | 99.53% | 99.90% | 0.00% |
| | | 0.0 | 87.05% | 98.20% | 100.00% | 100.00% | 20.67% | 31.59% | 100.00% | 99.97% | 0.03% |
| | | −0.05 | 83.59% | 28.13% | 30.16% | 28.16% | 20.67% | 8.16% | 92.08% | 93.81% | 0.03% |
| | **ResNeXt-50** | 0.1 | 95.27% | 94.81% | 100.00% | 100.00% | 6.29% | 11.68% | 98.84% | 99.17% | 0.00% |
| | | 0.0 | 87.85% | 96.84% | 100.00% | 100.00% | 18.88% | 28.56% | 100.00% | 99.90% | 0.03% |
| | | −0.05 | 84.79% | 23.20% | 24.40% | 23.34% | 20.07% | 9.02% | 95.27% | 96.74% | 0.03% |
| | **DenseNet-121** | 0.1 | 92.88% | 98.67% | 100.00% | 100.00% | 8.22% | 14.02% | 98.04% | 99.10% | 0.00% |
| | | 0.0 | 86.05% | 98.57% | 100.00% | 100.00% | 15.78% | 20.01% | 99.90% | 99.93% | 0.03% |
| | | −0.05 | 86.48% | 67.48% | 68.54% | 68.18% | 15.65% | 8.96% | 67.54% | 69.41% | 0.00% |

C.9 IMPACT OF LABEL SMOOTHING TRAINING ON BACKDOOR ATTACKS

In addition to adversarial robustness, we also investigate if training with LS has an impact on backdoor attacks. Backdoor attacks aim to integrate a secret behavior into a model that is only activated for inputs containing a pre-defined trigger (Gu et al., 2017; Struppek et al., 2023b). Due to the relatively small number of samples per class in FaceScrub and CelebA, we were not able to stably train the models – independently of the smoothing factor used. Therefore, we trained ResNet-152 models on poisoned ImageNette (Howard, 2019) datasets, which is a subset of ten ImageNet (Deng et al., 2009) classes. Specifically, we investigate the following common attack methods:

- **BadNets** (Gu et al., 2017): We added a $9 \times 9$ checkerboard pattern to the lower right corner of each image. In total, $10\%$ of all images were poisoned and labeled as class $0$.
- **Blended** (Chen et al., 2017): We interpolated each poisoned image with a fixed Gaussian noise pattern. The blend ratio was set to $0.1$. In total, $10\%$ of all images were poisoned and labeled as class $0$.

For evaluation, we computed the model's clean prediction accuracy on the test splits. The attack success is then measured by adding the triggers to all test images (excluding samples from the target class) and computed the share of poisoned samples that were classified as the target class $0$. The lower the attack success rate, the more robust the model is to an attack.

Table 12: Robustness of models against various backdoor attacks. We computed the attack success rates in all cases. The lower the success rate, the more robust a model is to the individual backdoor algorithm.

|  | $\alpha$ | Trigger | ↑ Clean Accuracy | ↓ Attack Success Rate |
|---|---|---|---|---|
| **ImageNette** | 0.1 | Clean | 81.71% | - |
|  |  | BadNets | 80.59% | 98.64% |
|  |  | Blended | 80.41% | 98.25% |
|  | 0.0 | Clean | 82.37% | - |
|  |  | BadNets | 81.17% | 100.0% |
|  |  | Blended | 80.31% | 95.48% |
|  | −0.05 | Clean | 78.17% | - |
|  |  | BadNets | 14.62% | 100.0% |
|  |  | Blended | 72.76% | 48.05% |

**Examples of Target Person**

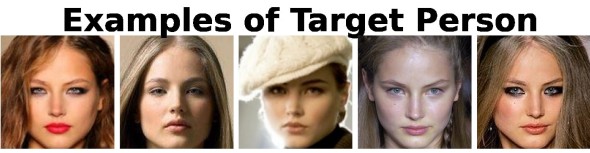

**Option 1**     **Option 2**     **Option 3**

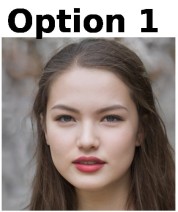 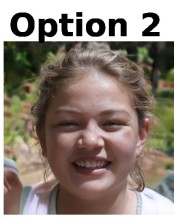 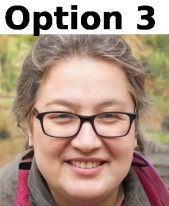

Figure 12: Labeling example from the user study. The annotators have to choose between the three options (MID, BiDO and negative LS) that look the least similar to the examples of the target person.

Table 13: PPA attack results against ResNet-152 models trained on FaceScrub and CelebA. The *User Preference* states the relative share of attack samples that annotators find to look the least similar to the target class.

| | FaceScrub | | | | | CelebA | | | | |
|---|---|---|---|---|---|---|---|---|---|---|
| Model | ↑ Test Acc | ↑ Acc@1 | ↓ $\delta_{face}$ | ↑ $\xi_{train}$ | User Preference | ↑ Test Acc | ↑ Acc@1 | ↓ $\delta_{face}$ | ↑ $\xi_{train}$ | User Preference |
| Neg. LS | 91.5% | 14.3% (-80.0) | 1.23 (+0.52) | 16.5% (-44.7) | 47.76% | 83.6% | 26.4% (-55.3) | 1.04 (+0.3) | 7.1% (-52.7) | 71.53% |
| MID | 91.1% | 92.0% (-2.3) | 0.72 (+0.01) | 73.0% (+11.8) | 17.16% | 80.4% | 78.0% (-3.8) | 0.74 (+0.0) | 70.9% (+11.1) | 10.22% |
| BiDO | 87.1% | 45.4% (-48.9) | 0.91 (+0.2) | 59.3% (-1.9) | 35.07% | 79.9% | 63.7% (-18.1) | 0.81 (+0.07) | 60.6% (+0.8) | 18.25% |

## C.10 USER-STUDY

In addition to the quantitative evaluation metrics used to evaluate the attacks' success, we conducted a qualitative user study to compare the effects of different defense methods on the visual similarity between attack results and the true identity. We compared the attack results on the ResNet-152 FaceScrub and CelebA models trained with negative LS ($\alpha = -0.05$) to the attack results against models trained with MID and BiDO, respectively. The models are the same as stated in Tab. 1. The study was conducted using the manual labeling service of thehive.ai. More specifically, we randomly sampled 150 target classes from the FaceScrub and CelebA datasets, respectively. We then took for each target model and target class the attack results that achieved the highest robustness on the target model using PPA's final selection algorithm. For each class, the annotators were presented with 5 randomly selected images of the true identity and the 3 attack results from the attack against negative LS, MID, and BiDO models. The annotators were then asked to "select the person (options 1-3) that looks the least similar to the person depicted in the example images". Noticeably, the annotators were not provided any information on the different defense mechanisms or even the attack setting at all to avoid undesired side effects during the labeling process. Fig. 12 shows an example from the user study. Each sample was labeled by three different annotators and the final decision was made on a majority vote. For 16 FaceScrub samples and 13 CelebA samples, the annotators could not decide on a single favorite. We excluded those samples from the final preference computation. In total, 37 annotators took part in the labeling process. To filter out unreliable annotators, some honeypot tasks were added to ensure the annotation quality.

Tab. 13 states the user study results as the relative preference for the individual defense mechanisms. For the FaceScrub model, the annotators found about 48% of the attack results from attacks against the negative LS model to look the least similar to the target class. For the CelebA experiment, even 71.5% of the results were preferred. Overall, this means that the defensive effect of negative LS was qualitatively higher valued than MID and BiDO as defense mechanisms, particularly on the CelebA models. At the same time, negative LS also maintained the model's utility in terms of test accuracy better than the other defense mechanisms.

