# OpenReview forum: "Be Careful What You Smooth For: Label Smoothing Can Be a Privacy Shield but Also a Catalyst for Model Inversion Attacks"
_ICLR.cc/2024/Conference — ICLR 2024 poster_

### Official Review · Reviewer_EAeB · 2023-10-26

**Soundness:** 3 good
**Presentation:** 3 good
**Contribution:** 2 fair
**Rating:** 5
**Confidence:** 4

**Summary:**

The manuscript analyzes the potential risk of increased privacy leakage associated with traditional positive Label Smoothing in the context of Model Inversion Attacks. Additionally, it provides an analysis on how utilizing negative values can counter this risk.

**Strengths:**

The authors have for the first time considered the potential privacy leakage risks associated with the use of positive label smoothing under model inversion attacks, and have provided insights into the privacy benefits that may arise from using negative label smoothing. Additionally, the authors have presented a framework for analyzing the exposure of privacy in model inversion attacks due to label smoothing, and have examined how label smoothing may affect the sampling and optimization phases of model inversion attacks. Furthermore, the authors have provided a geometric intuition in the EMBEDDING SPACES to explain this phenomenon, all of which are relatively novel contributions. The quality of writing in the article is excellent, the presentation is clear, the experiments are comprehensive, and the work holds significant importance.

**Weaknesses:**

1. The paper lacks a theoretical analysis framework. The authors claim that negative label smoothing performs better than state-of-the-art defenses, but in fact, it only performs better under model inversion attacks within the analysis framework of this paper. However, privacy protection mechanisms are often capable of resisting multiple types of attacks (or even arbitrary attacks), and it is still unknown whether the approach of negative label smoothing is effective in protecting privacy against other types of attacks.

2. It is generally considered that achieving both privacy and efficiency in Euclidean space through privacy protection mechanisms is an NP-hard problem. The paper does not discuss whether the performance improvement of using negative label smoothing is significantly less than that of positive label smoothing. It is noted that when negative label smoothing was previously proposed, the range of parameter choices was from negative infinity to 1. In this case, good performance does not mean that always choosing negative label smoothing will result in good performance. The authors do not discuss in the article how the performance of using negative label smoothing compares to that of positive LS or even not using this regularization measure at all, as well as whether negative LS is not applicable in many tasks.

**Questions:**

1. Can an analysis be provided comparing the performance of using negative label smoothing to the performance under positive label smoothing?

2. Are there any known vulnerabilities introduced by using negative label smoothing?

3. How generalizable are the findings of this paper? Are the observed effects of label smoothing on privacy specific to the datasets and models used in the experiments, or can they be applied to other domains and architectures as well?

**Details Of Ethics Concerns:**

This article might inspire people to use model inversion attacks to target some existing models trained with positive label smoothing.

---

> ### Author Response · Authors · 2023-11-13
> **clarification of term "performance"**
>
> Dear Reviewer EAeB,
>
> thank you for your detailed review. We are currently working on the rebuttal to answer your questions and provide additional results. To provide a satisfying answer for you, we would like to ask for a more clear definition, of what you mean by "good performance" as in "Can an analysis be provided comparing the performance of using negative label smoothing to the performance under positive label smoothing?".
>
> Do you mean a model's prediction performance on an unseen dataset (classification test accuracy)? If not, a clarification of the term "performance" would make it easier for us to prepare our answer.
>
> Best regards,
> the authors

---

> > ### Author Response · Authors · 2023-11-15
> >
> > Dear Reviewer EAeB,
> > thank you very much for reviewing our paper. We are glad you find our analyses novel and important, the writing excellent and the experiments comprehensive. We try to address all your remarks and questions sufficiently.
> >
> > **1.) Lack of theoretical analysis framework:** Providing a grounded theoretical framework for our work already requires a universally accepted theoretical framework for model inversion attacks. Unfortunately, to date, there is no such framework available, and since sensitive features and privacy leakage themselves are hard to clearly define, this is no trivial task. We also stress that previous attack and defense methods all also not provide a theoretical framework. However, we believe that our extensive experiments and analyses already provide proof for our statements.
> >
> > **2.) Privacy protection mechanism against other types of attacks:** We kindly emphasize that our focus lies only on model inversion attacks. We already provided an experimental evaluation on the impact of label smoothing on other model inversion algorithms besides PPA in **Appx. C.5.**. Moreover, as requested by other reviewers, we also added empirical results for adversarial attacks (see **Appx. C.8** or our answer to Reviewers uLpV and wJ7G). For other privacy attacks besides model inversion attacks, we cannot make any guarantees, e.g., differential privacy. For such a privacy guarantee, the community is still missing a clear definition of privacy leakage in the feature space. There exists no clear definition of when we can call an inversion attack successful, or at which point a person's identity is leaked from a reconstructed sample. That is why existing attacks try to approximate such statements with metrics like the attack accuracy, the FaceNet distance, or our novel knowledge extraction score.
> >
> > **3.) Can an analysis be provided comparing the performance of using negative label smoothing to the performance under positive label smoothing?** We are not sure if we understand your point here correctly. We state for all models their classification performance as the prediction accuracy on a hold-out test set, for example, in **Table 1 and Table 2**. In addition, we state each model's calibration error (ECE) and a total of 7 metrics for quantifying the attack performance of model inversion attacks. If we misunderstand your definition of performance, please let us know how we can provide you with the desired information.
> >
> > **4.) Are there any known vulnerabilities introduced by using negative label smoothing?** Not directly a vulnerability but a side effect of using negative label smoothing is the loss of model calibration, i.e., prediction scores matching the true probability of a predictions correctness. Moreover, we added several experiments to investigate the impact of label smoothing on adversarial examples (see **Appx. C.8**). In case of adversarial robustness, our results show that negative label smoothing can even increase a models robustness, making it less susceptible to image manipulations.
> >
> > **5.) How generalizable are the findings of this paper?** Please note that we already conducted experiments on different datasets (FaceScrub and CelebA), as well as on different common model architectures (ResNet, ResNeXt, DenseNet, VGG16). So we can already conclude that the effect is not limited to a single dataset and model architecture. Since label smoothing is applicable to all kinds of classification tasks and not dependent on a specific domain, we expect our results to be also valid for other domains and model architectures.

---

> > > ### Author Response · Authors · 2023-11-16
> > >
> > > Dear Reviewer EAeB,
> > > to provide additional empirical proof to your question, if there are any known vulnerabilities introduced by using negative label smoothing, we just added additional results for the impact of backdoor attacks. Our results show that negative label smoothing responds sensitively to the poisoned datasets, which results in a reduction in clean accuracy and, in the case of blended backdoors, a significantly reduced attack success rate. As a result, negative label smoothing may serve as a proxy for dataset quality as it reduces clean model accuracy, which may indicate that the dataset has been tampered with. But seems to not introduce an additional vulnerability for backdoors.
> > >
> > > For more details and hyperparameters, please take a look at **Appx. C.9**.

---

### Official Review · Reviewer_uLpV · 2023-10-28

**Soundness:** 2 fair
**Presentation:** 3 good
**Contribution:** 3 good
**Rating:** 8
**Confidence:** 4

**Summary:**

This paper studies how label smoothing (during model training) can affect the performance of the model inversion attacks.
Specifically, the authors find that a positive label smoothing factor would facilitate the inversion attacks, while a negative factor would suppress the attacks.
This phenomenon underlines the importance of delving more into factors that influence machine learning privacy leakage.

**Strengths:**

1. This paper reveals a novel phenomenon that the positivity/negativity of label smoothing factors can affect privacy leakage in model inversion attacks. Actually, the mechanism behind this can also be explained as the robustness-privacy trade-off: [r1] first find that label smoothing with positive factor can improve adversarial robustness, and [r2] first find that there is a trade-off between adversarial robustness and membership inference attacks. Nevertheless, although the relationship between label smoothing and model inversion attacks is forecastable based on the aforementioned works, this paper is the first that empirically demonstrates the relationship. As a result, I think the results of this paper are fundamental and important.

2. The paper is easy to follow and provides sufficient experiments details for reproducing its results.


**References:**

[r1] Shafahi et al. "Label smoothing and logit squeezing: A replacement for adversarial training?" arXiv 2019.

[r2] Song et al. "Privacy risks of securing machine learning models against adversarial examples". CCS 2019.

**Weaknesses:**

1. In Section 3 and Figure 1, the authors provide an intuitive explanation of why label smoothing can affect model inversion attacks based on a simple experiment on 2D data. The explanation is based on the hypothesis that a more clear decision boundary would make training data less likely to be leaked through model inversion attacks. This hypothesis is odd (at least for me) and I think the authors may need to put more effort into explaining why the hypothesis makes sense.


2. Negative-factor label smoothing would result in the smoothed label no longer a probability simplex. I think the authors may need to justify why this type of label smoothing is appropriate.


3. Suggestion: As explained in Section "Strengths", the found phenomenon can be seen as a robustness-privacy trade-off. Since this paper finds that negative-factor label smoothing can mitigate privacy leakage by model inversion attacks, I suspect it could also harm the adversarial robustness of the model. Therefore, I suggest the authors include a discussion on the potential harm to models' robustness when protecting training data privacy.

**Questions:**

None.

---

> ### Author Response · Authors · 2023-11-15
>
> Dear Reviewer uLpV,
> thank you very much for reviewing our paper. We are happy that you think our presented results are fundamental and important. We try to address all your remarks appropriately.
>
> **1.) Why would a clear decision boundary make training data less likely to be leaked?** We think there might be a misunderstanding here. Figure 1 indeed visualizes the decision boundaries (which are quite similar for all three models) together with the classifiers' confidence scores for each point in the input space. However, the paper's argumentation is not that negative label smoothing (Fig. 1c) defends against model inversion attacks because the model has clear decision boundaries. We rather argue that negative label smoothing leads to models that provide no clear guidance signal for the model inversion attack algorithm. In this toy example, the model predicts high confidence scores for almost all inputs, excepts those close to the decision boundary. Therefore, not much information about the positioning of the training distribution in the input space is leaked, and the attack is not able to move close towards the training data. Positive label smoothing, on the other hand, places the training data in a high confidence area, therefore, making it easier for the attack algorithm to guide towards this position in the input space.
>
> **2.) Negative label smoothing creates no valid probability simplex:** Indeed, the labels smoothed with a negative factor define no valid probability distribution due to the negative probabilities (if the labels were interpreted as probabilities). We already brought this point up in Section 2.2, and it was already discussed by Wei et al. [1], who originally proposed negative label smoothing for tackling label noise. The interesting effect arises from putting negatively smoothed labels into the cross-entropy loss, which avoids a gradient saturation effect during training. Consequently, negative label smoothing gives the model an incentive to increase its logit output for the true label while reducing the logits for all other classes. We already provide a more comprehensive, formal analysis of label smoothing with positive and negative smoothing factors in **Appx. A**.
>
> [1] Wei et al., To Smooth or Not? When Label Smoothing Meets Noisy Labels

---

> > ### Author Response · Authors · 2023-11-15
> >
> > **3.) Include a discussion on the potential harm to models' robustness when protecting training data privacy:** We added additional analyses on the impact of label smoothing on a model's robustness to adversarial examples (**Appx. C.8**). Results on backdoor attacks will be added as soon as they become available.
> >
> > For adversarial attacks, we investigated four attack algorithms (FGSM, PGD, BIM, One-Pixel Attack) in two modes (targeted + untargeted) to measure the impact of label smoothing on a model's robustness. Please refer to **Appx. C.8** for the detailed hyperparameters and attack results. Overall, we found that training with positive label smoothing makes the models slightly more robust to adversarial examples. Interestingly, training with negative label smoothing markedly increased the models' robustness to adversarial attacks. So, as a first result in this research direction, we can conclude that negative label smoothing not only helps mitigate privacy leakage but can also make a model more robust to adversarial examples. We will further investigate this phenomenon in future work.
> >
> > *Columns 3-6 state the attack success rate for untargeted adversarial attacks, columns 7-10 the success rate for targeted attacks.*
> >
> > **FaceScrub**
> > | α| ↑ Clean Test Acc | ↓ FGSM | ↓ PGM |↓ BIM |↓ One-Pixel |↓ FGSM |↓ PGM | ↓ BIM | ↓ One-Pixel|
> > | ------- |-------- | ------- |-------- | ------- |-------- | ------- |-------- | ------- |-------- |
> > |ResNet-152|
> > |0.1 |97.39%|70.09%| 100.00% |100.00%| 3.93%| 7.76% |92.42% |93.51%| 0.00%|
> > |0.0| 94.93% |96.09%| 100.00% |100.00%| 7.63%| 55.62%| 99.95% |99.97% |0.03%|
> > |−0.05| 91.45% |13.38% |13.94% |13.44% |10.43% |22.60% |98.52%| 98.89%| 0.08%|
> > |ResNeXt-50|
> > |0.1| 97.54%| 59.21% |100.00% |100.00% |3.54% |6.84% |83.58%| 85.56%| 0.03%|
> > |0.0| 95.25% |97.36%| 100.00%| 100.00%| 7.79% |52.46% |100.00%| 99.97%| 0.05%|
> > |−0.05 |92.40% |11.17% |11.38%| 11.11% |9.74% |24.00%| 99.55% |99.74%| 0.08%|
> > |DenseNet-121|
> > |0.1| 97.15%| 81.76%| 100.00%| 100.00% |3.83%| 9.16% |93.11%| 94.59% |0.03%|
> > |0.0 |95.72%| 97.02%| 100.00%| 100.00% |5.44%| 39.73%| 100.00%| 100.00%| 0.05%|
> > |−0.05| 92.13% |31.84% |32.74%| 32.15%| 9.61%| 18.72% |86.17% |88.52%| 0.08%|
> >
> >
> >
> > **CelebA**
> > | α| ↑ Clean Test Acc | ↓ FGSM | ↓ PGM |↓ BIM |↓ One-Pixel |↓ FGSM |↓ PGM | ↓ BIM | ↓ One-Pixel|
> > | ------- |-------- | ------- |-------- | ------- |-------- | ------- |-------- | ------- |-------- |
> > |ResNet-152|
> > |0.1| 95.11%| 97.44% |100.00%| 100.00% |6.49%| 17.64%| 99.53%| 99.90% |0.00%|
> > |0.0| 87.05%| 98.20% |100.00%| 100.00%| 20.67% |31.59%| 100.00%| 99.97%| 0.03%|
> > |−0.05| 83.59% |28.13%| 30.16% |28.16%| 20.67%| 8.16%| 92.08% |93.81%| 0.03%|
> > |ResNeXt-50|
> > |0.1 |95.27% |94.81% |100.00% |100.00% |6.29% |11.68% |98.84% |99.17% |0.00%|
> > |0.0 |87.85% |96.84% |100.00%| 100.00% |18.88%| 28.56% |100.00% |99.90% |0.03%|
> > |−0.05| 84.79%| 23.20%| 24.40% |23.34% |20.07% |9.02%| 95.27%| 96.74% |0.03%|
> > |DenseNet-121|
> > |0.1 |92.88% |98.67%| 100.00% |100.00%| 8.22%| 14.02% |98.04%| 99.10% |0.00%|
> > |0.0 |86.05%| 98.57%| 100.00% |100.00%| 15.78% |20.01%| 99.90% |99.93% |0.03%|
> > |−0.05| 86.48% |67.48%| 68.54%| 68.18% |15.65%| 8.96%| 67.54% |69.41%| 0.00%|

---

> ### Comment · Reviewer_uLpV · 2023-11-19
> **Score has been raised**
>
> Thanks to the authors for their detailed respone.
> Given that all my concerns have been addressed, I have raised my score to 8.

---

### Official Review · Reviewer_wJ7G · 2023-10-30

**Soundness:** 3 good
**Presentation:** 3 good
**Contribution:** 3 good
**Rating:** 6
**Confidence:** 4

**Summary:**

This paper studies the impact of label smoothing, a commonly used regularization techniques in deep learning, on the privacy vulnerability of models to model inversion attacks (MIAs), which aim to reconstruct the characteristic features of each class. It is shown that traditional label smoothing with positive factors may inadvertently aid MIAs, increasing the privacy leakage of a model. The paper also finds that smoothing with negative factors can counteract this trend, impeding the extraction of class-related information.

**Strengths:**

- MIAs represent a major threat to the privacy of machine learning models. This seems to be the first paper studying the relationship between label smoothing and MIAs.
- The work provides both empirical and analytical justification for the connections between label smoothing and MIAs.
- The ablation study is interesting to show that label smoothing impacts MIAs mainly in the optimization stage.

**Weaknesses:**

- The findings are not very surprising. Given that label smoothing helps the model generalize, leading to better representation of each class (e.g., more smooth decision boundaries). It is intuitive that more smooth decision boundaries allow gradient-based method to better optimize the representation of each class.
- There is trade-off between the model's accuracy and its vulnerability to MIAs. Label smoothing with negative factors reduces the privacy risks at the cost of model accuracy (see Table 1). Intuitively, a poorly trained model is more robust to MIAs.

**Questions:**

How does label smoothing impact the model's vulnerability to other attacks (e.g., adversarial attacks, backdoor attacks, etc)?

---

> ### Author Response · Authors · 2023-11-15
>
> Dear Reviewer  wJ7G,
> thank you very much for reviewing our paper. We are glad that you find our paper tackling an interesting and important problem. We try to address all your remarks and questions.
>
> **1.) Findings are not very surprising:** Please excuse that we kindly disagree here. To our knowledge, no previous research has investigated the impact of label smoothing on a model's privacy leakage or even discussed such a possibility when training with positive label smoothing. Particularly the effects of negative label smoothing have generally been barely investigated, and its impact on a model's privacy is important to examine. For people familiar with the standard label smoothing literature, some of our findings might be intuitive (which is not a bad thing at all), but the general community seems not to be aware of its effects on a model's privacy. We also think that the impact of negative label smoothing is not trivial to see, even for people familiar with the topic. We believe that a comprehensive study on the impact of label smoothing on a model's privacy leakage under the lens of model inversion is critical and very relevant to the privacy research community and will inspire future efforts in this direction.
>
> **2.) Accuracy-vulnerability trade-off:** While there definitely exists an inherent trade-off between a model being able to recognize certain features in inputs required to make good predictions and its privacy leakage, we stress that the slightly reduced prediction accuracy of the models trained with negative LS is not the main reason for their defensive effects. To demonstrate this effect, we re-trained the models with positive LS and standard cross-entropy loss and stopped each training run as soon as the model's test accuracy matched that of the model trained with negative LS. We then repeated the attacks against these models, which all achieve comparable prediction accuracy, to see if the attack success of the models drops. Our results demonstrate that even under comparable test accuracy, the attacks still perform well and substantially better than on the negative LS models. This clearly indicates that reducing a model's prediction accuracy is not the driving factor behind negative LS as a defense mechanism. This can also be seen when comparing with BiDO and MID, which both achieve lower test accuracy than negative label smoothing but still are more vulnerable to the attack.
>
> |  FaceScrub |  α |  ↑ Test Acc |  ↑ Acc@1 | ↑ Acc@5 | ↓ δface | ↓ δeval |  ↓FID |
> | -------- | ------- |------- |------- |------- |------- |------- |------- |
> No Label Smoothing |0.0 |91.74%| 89.68%| 99.00% |0.7382 |133.12 |50.43|
> Positive Label Smoothing| 0.1 |92.13%| 87.16% |97.20% |0.7766 |135.73| 49.56|
> Negative Label Smoothing |−0.05| 91.45% |14.34%| 30.94% |1.2320 |239.02 |59.38 |16.45% |15.73%|
>
> |  CelebA |  α |  ↑ Test Acc |  ↑ Acc@1 | ↑ Acc@5 | ↓ δface | ↓ δeval |  ↓FID |
> | -------- | ------- |------- |------- |------- |------- |------- |------- |
> |No Label Smoothing |0.0 |84.02%| 69.36% |90.60%| 0.7899| 344.82| 49.60|
> |Positive Label Smoothing |0.1 |86.78% |76.64%| 92.84%| 0.785| 332.53| 50.05|
> |Negative Label Smoothing |−0.05 |83.59%| 26.41%| 49.96%| 1.0420| 441.67 |61.30 |7.08%| 5.89%|
>
> Our results including a more detailed experimental setting and more elaborated discussion are also stated in **Appendix C.6**.

---

> ### Author Response · Authors · 2023-11-15
>
> **3.) How does label smoothing impact the model's vulnerability to adversarial attacks and backdoor attacks?** We stress that the paper focuses on the domain of model inversion attacks, and adversarial examples and backdoor attacks are out of the general scope of the paper. However, we agree that the impact of label smoothing on security attacks is indeed a very interesting question. We conducted various additional experiments in the setting of adversarial attacks to provide additional insights.
>
> For adversarial attacks, we used four attack algorithms (FGSM, PGD, BIM, One-Pixel Attack) in two modes (targeted + untargeted) to measure the robustness of models trained with hard labels, positive label smoothing, or negative label smoothing. Please refer to **Appendix C.8** for the detailed hyperparameters and attack results. Overall, we found that training with positive label smoothing makes the models slightly more robust to adversarial examples. Interestingly, training with negative label smoothing markedly increased the models' robustness to adversarial attacks. So, as a first result in this research direction, we can conclude that negative label smoothing not only helps mitigate privacy leakage but can also make a model more robust to adversarial examples. We will look further into this direction of adversarial attacks in future work.
>
> *Columns 4-7 state the attack success rate for untargeted adversarial attacks, columns 8-11 the success rate for targeted attacks.*
>
> *Columns 3-6 state the attack success rate for untargeted adversarial attacks, columns 7-10 the success rate for targeted attacks.*
>
> **FaceScrub**
> | α| ↑ Clean Test Acc | ↓ FGSM | ↓ PGM |↓ BIM |↓ One-Pixel |↓ FGSM |↓ PGM | ↓ BIM | ↓ One-Pixel|
> | ------- |-------- | ------- |-------- | ------- |-------- | ------- |-------- | ------- |-------- |
> |ResNet-152|
> |0.1 |97.39%|70.09%| 100.00% |100.00%| 3.93%| 7.76% |92.42% |93.51%| 0.00%|
> |0.0| 94.93% |96.09%| 100.00% |100.00%| 7.63%| 55.62%| 99.95% |99.97% |0.03%|
> |−0.05| 91.45% |13.38% |13.94% |13.44% |10.43% |22.60% |98.52%| 98.89%| 0.08%|
> |ResNeXt-50|
> |0.1| 97.54%| 59.21% |100.00% |100.00% |3.54% |6.84% |83.58%| 85.56%| 0.03%|
> |0.0| 95.25% |97.36%| 100.00%| 100.00%| 7.79% |52.46% |100.00%| 99.97%| 0.05%|
> |−0.05 |92.40% |11.17% |11.38%| 11.11% |9.74% |24.00%| 99.55% |99.74%| 0.08%|
> |DenseNet-121|
> |0.1| 97.15%| 81.76%| 100.00%| 100.00% |3.83%| 9.16% |93.11%| 94.59% |0.03%|
> |0.0 |95.72%| 97.02%| 100.00%| 100.00% |5.44%| 39.73%| 100.00%| 100.00%| 0.05%|
> |−0.05| 92.13% |31.84% |32.74%| 32.15%| 9.61%| 18.72% |86.17% |88.52%| 0.08%|
>
>
>
> **CelebA**
> | α| ↑ Clean Test Acc | ↓ FGSM | ↓ PGM |↓ BIM |↓ One-Pixel |↓ FGSM |↓ PGM | ↓ BIM | ↓ One-Pixel|
> | ------- |-------- | ------- |-------- | ------- |-------- | ------- |-------- | ------- |-------- |
> |ResNet-152|
> |0.1| 95.11%| 97.44% |100.00%| 100.00% |6.49%| 17.64%| 99.53%| 99.90% |0.00%|
> |0.0| 87.05%| 98.20% |100.00%| 100.00%| 20.67% |31.59%| 100.00%| 99.97%| 0.03%|
> |−0.05| 83.59% |28.13%| 30.16% |28.16%| 20.67%| 8.16%| 92.08% |93.81%| 0.03%|
> |ResNeXt-50|
> |0.1 |95.27% |94.81% |100.00% |100.00% |6.29% |11.68% |98.84% |99.17% |0.00%|
> |0.0 |87.85% |96.84% |100.00%| 100.00% |18.88%| 28.56% |100.00% |99.90% |0.03%|
> |−0.05| 84.79%| 23.20%| 24.40% |23.34% |20.07% |9.02%| 95.27%| 96.74% |0.03%|
> |DenseNet-121|
> |0.1 |92.88% |98.67%| 100.00% |100.00%| 8.22%| 14.02% |98.04%| 99.10% |0.00%|
> |0.0 |86.05%| 98.57%| 100.00% |100.00%| 15.78% |20.01%| 99.90% |99.93% |0.03%|
> |−0.05| 86.48% |67.48%| 68.54%| 68.18% |15.65%| 8.96%| 67.54% |69.41%| 0.00%|
>
> Results  including a more detailed experimental setting and more elaborated discussion are also stated in Appendix **C.8**.
>
> Experiments investigating the backdoor setting are currently being implemented. We will provide the results as soon as they become available.

---

> > ### Author Response · Authors · 2023-11-16
> >
> > Dear Reviewer wJ7G,
> > we just added the results for the backdoor experiments. Due to the relatively small number of samples per class in FaceScrub and CelebA, we were not able to stably train the models - independently of the smoothing factor used. Therefore, we trained ResNet-152 models on poisoned ImageNette datasets, which is a subset of ten ImageNet classes. Specifically, we investigated the common BadNets and Blended attacks.
> >
> > Overall, positive label smoothing seems not to make a model more vulnerable to backdoor attacks; training with standard cross-entropy loss already builds a strong connection with the attacks' trigger. Negative label smoothing, however, reacts sensitively to the poisoned datasets, leading to decreased clean accuracy and, in the case of Blended backdoors, also a substantially lower attack success rate. So negative label smoothing can act as some kind of indicator of the dataset quality since it lowers the clean model accuracy, which might be a sign that the dataset is somehow manipulated.
> >
> > | α |Trigger |↑ Clean Accuracy |↓ Attack Success Rate|
> > | -------- | ------- |------- |------- |
> > | 0.1|
> > ||Clean |81.71%| -|
> > ||BadNets| 80.59%| 98.64%|
> > ||Blended| 80.41% |98.25%|
> > |0.0|
> > ||Clean |82.37%| -|
> > ||BadNets| 81.17% |100.0%|
> > ||Blended |80.31% |95.48%|
> > |−0.05|
> > ||Clean |78.17%| -|
> > ||BadNets |14.62%| 100.0%|
> > ||Blended |72.76% |48.05%|
> >
> > For more details and hyperparameters, please take a look at **Appx. C.9**.

---

> > > ### Comment · Reviewer_wJ7G · 2023-11-22
> > >
> > > I thank the authors for the response and added experiments, which have addressed some of my questions. My concern about the interestingness of the findings remains. I'm inclined to increase my score and will do so after discussing with fellow reviewers and AC.

---

> > > > ### Author Response · Authors · 2023-11-23
> > > >
> > > > Dear Reviewer wj7G,
> > > >
> > > > thank you for reading our rebuttal and your answer. We are happy that you are inclined to increase your score. If there is anything still missing for you or if you require any additional content (for the final paper since the rebuttal phase is ending), we are happy to work on adding it to the paper.
> > > >
> > > > Best,
> > > > the authors

---

### Official Review · Reviewer_dNRe · 2023-10-31

**Soundness:** 2 fair
**Presentation:** 3 good
**Contribution:** 2 fair
**Rating:** 6
**Confidence:** 5

**Summary:**

This work considers the problem of defending against Model Inversion Attacks in white-box settings. The major contributions of this work are:

1) In the context of model inversion attacks, this paper observes that positive label smoothing increases a model’s privacy leakage and negative label smoothing counteracts such effects.

2) Consequently, negative label smoothing is proposed as a proactive measure to defend against Model Inversion Attacks.

**Strengths:**

1) This paper is written well and it is easy to follow.

2) To my knowledge, this is the first work to explore label smoothing in the context of defense against model inversion attacks.

**Weaknesses:**

At a high level, Model Inversion attack procedures can be decomposed into 2 stages: 1) Classifier training, and 2) Model Inversion attack on the classifier. My review below addresses weaknesses of this work corresponding to each stage.

**Classifier training:**

1) Section 3. Analysis is only remotely related to contemporary Model Inversion setups. In particular, it is unclear as to how observation from Section 3 lays groundwork to the remainder of the paper.

- What is the optimization objective for both classifier training and model inversion attack for Figure 1? A clearly formulated problem definition with relevant equations is critical to understand this part.\

- How would Figure 1 change if the number of iterations = 5K for all setups.

2) Section 4.3. A large number of observations from Sec 4.3 for Standard and Positive Label Smoothing has already been thoroughly investigated in prior works.

- Muller et al. [A] and Chandrasegaran et al. [B] have already shown that positive label smoothing erases some relative information in the logits resulting in better class-wise separation of penultimate layer representations under positive LS compared standard training.

- Figure 4, column 3 is unclear. What are the Training and Test accuracies of the classifiers used in Figure 4?  Recall that if the classifier trained with negative label smoothing is good, penultimate layer representations should be linearly separable. Therefore, does column 3 correspond to a poor classifier trained with negative label smoothing?

3) Can the authors explain why standard training is required in the first few iterations before “gradually” increasing the negative label smoothing?



**Model Inversion Attacks:**

1) Limited empirical study. This work only studies one attack method for evaluating defense (No comparison against MID and BiDO defense setups even in Table 8 in Supp.). I agree that PPA works well in high-resolution setups, but SOTA attacks in well established test beds are required to understand the efficacy of the proposed defense.

- It is important to include GMI, KEDMI, VMI [C], LOMMA [4] and PLG-MI [5] attacks to study the efficacy of the proposed defense (against other SOTA defense methods). Currently it is not possible to compare the proposed method with results reported in the MID and BiDO defense papers.

2) There is no evidence (both qualitative and quantitative) to establish that unstable gradient directions during Model Inversion attack is due to negative label smoothing. Is it possible that such shortcomings could be due to the PPA attack optimization objectives? Addressing 1) above can answer this question to some extent.

3) User studies are necessary to show the defense/ leakage of privacy shown by the inversion results. Since this work focuses on private data reconstruction, it is important to conduct user study to understand the improvements (See [F]).

4) Significant compromise in model utility when using negative label smoothing questioning the findings/ applicability of this approach. Table 1 and Table 8 results suggest that Neg. LS reduces the Model Accuracy (model utility) by huge amounts, i.e.: A 3.5% reduction in Test Accuracy for CelebA (Table 1) compared to Standard training could be serious. Recall that lower model accuracy leads to lower MI attack results. I agree that generally some compromise in model utility might be required for defense, but large reduction in model utility makes this approach questionable, i.e., In practice, no one would deploy/ attack a weaker model.

5) Error bars/ Standard deviation for experiments are missing.

6) Missing related works [C, D, E].


Overall I enjoyed reading this paper. But in my opinion, the weaknesses of this paper outweigh the strengths. But I’m willing to change my opinion based on the rebuttal.

===

[A] Müller, Rafael, Simon Kornblith, and Geoffrey E. Hinton. "When does label smoothing help?." Advances in neural information processing systems 32 (2019).

[B] Chandrasegaran, Keshigeyan, et al. "Revisiting Label Smoothing and Knowledge Distillation Compatibility: What was Missing?." International Conference on Machine Learning. PMLR, 2022.

[C] Wang, Kuan-Chieh, et al. "Variational model inversion attacks." Advances in Neural Information Processing Systems 34 (2021): 9706-9719.

[D] Nguyen, Ngoc-Bao, et al. "Re-thinking Model Inversion Attacks Against Deep Neural Networks." Proceedings of the IEEE/CVF Conference on Computer Vision and Pattern Recognition. 2023.

[E] Yuan, Xiaojian, et al. "Pseudo Label-Guided Model Inversion Attack via Conditional Generative Adversarial Network." AAAI 2023 (2023).

[F] [MIRROR] An, Shengwei et al. MIRROR: Model Inversion for Deep Learning Network with High Fidelity. Proceedings of the 29th Network and Distributed System Security Symposium.


=======================================================

Post-rebuttal

Thank you authors for the extensive rebuttal. I've adjusted my rating to 6 to acknowledge your efforts, though I still have reservations about the analytical support for your claims. I'd like to offer some final thoughts for consideration:

- **Clarifying the paper's focus primarily on white-box attacks could be beneficial.**  It may enhance the paper to discuss how the findings, particularly regarding negative label smoothing's impact on optimization, apply to black-box and label-only model inversion attacks that employ gradient-free approaches (Table 8).

- Additionally, it would be great to mention a short description reg. any anomalous results in the captions of relevant tables.

I appreciate your extensive efforts and involvement during the rebuttal phase.

**Questions:**

Please see Weaknesses section above for a list of all questions.

---

> ### Author Response · Authors · 2023-11-15
>
> Dear Reviewer dNRe,
> thank you very much for reviewing our paper. We are happy that you enjoyed reading our paper. We hope we can clear up any ambiguities, provide the requested details and results, and convince you more of our research.
>
> ### Classifier Training
>
> **1.) Additional Details for Figure 1:** To provide more details on the hyperparameters and optimization goals in Figure 1, we added Section B.6. to the Appendix. Specifically, each model was trained with a cross-entropy loss, as defined by Eq. (3) in the main paper. Each training only differs in the label smoothing factor used ($\alpha \in \{0, 0.05, -0.05\}$). So the classifier training does conceptually not differ from the training of the face recognition models. For the attack, we used the identity loss proposed by Zhang et al.[1], which corresponds to a simple cross-entropy objective. Since no image prior is used, there is no additional prior loss. Optimization itself was done using standard SGD. The attack goal is to adjust the features of the initial sample from the green circles class to recover features from the orange pentagons class. This setting simulates a simple model inversion attack with the goal to recover features from the target class, i.e., in the simple 2D setting moving the sample as close as possible to the target distribution. For more detailed descriptions and parameters, please have a look at **Appendix B.6**.
>
> As requested, we also repeated the experiment and set the number of optimization steps to 5,000 without any stopping mechanism applied. The result, which we state in **Figure 7 in Appx. B.6**, draws a similar picture. Again, the attack against the model trained with positive LS moves the optimized sample close to the target distribution's center, whereas the optimization on the model trained without LS moves closer to the target distribution but stays still on the surface of it. For the negative LS model, not much has changed. The optimized sample moves a bit further away from the decision boundary but still is far away from the training distribution.
>
> Overall, the toy example motivates our research into the effect of label smoothing on model inversion attacks. Particularly, the setting shows that models trained with positive LS provide better guidance signals for model inversion attacks, leading the optimization closer to the true training distribution, whereas training with negative smoothing complicates the optimization process and hinders the recovery of sensitive features. These effects are then later shown on face recognition models, for which the FaceNet distance might be the most important metric to demonstrate that attacks against positive LS models recover more features of the target distribution, whereas attacks against negative LS models revert this effect.

---

> > ### Author Response · Authors · 2023-11-15
> >
> > **2.) Observations are partly known in literature:** The stated previous literature investigated the impact of positive label smoothing on knowledge distillation and showed that this can result in better class-wise separation of penultimate layer representations. In this work, however, we look at the information leakage by model inversion attacks, a completely different setting since the attack goal is not to train a student model but to reconstruct sensitive features of individual classes Also, compared to previous work, we also incorporate an analysis of negative label smoothing. However, we added references to previous research in Section 4.3 to place our findings better within the existing literature.
> >
> > **3.) Training and Test Accuracy of Classifiers in Fig. 4?** As stated in the paper (*"In the following, we focus our analyses on the FaceScrub models..."*), all analyses are based on the FaceScrub ResNet-152 models, which are stated in **Table 1** and, with more details, in **Table 2 in Appendix C.1**. However, we understand that this information might easily be overlooked, so we made it more clear in the paper. Regarding the training and test accuracy, the classifiers achieved the following results:
> > Consequently, the model used for Figure 4c) is no poorly trained classifier but the ResNet-152 FaceScrub model achieves generally good performance on the test data.
> >
> > | Smoothing Factor |  Test Accuracy|  Training Accuracy|
> > | -------- | ------- | ------- |
> > | α = 0.1 | 97.39%|  99.99%|
> > | α = 0.0 | 94.93%|  99.99%|
> > | α = −0.05 | 91.45% | 97.63%|
> >
> > **4.) Why use a warm-up period when training with negative LS?** We follow here the insights of Wei et al. [2]. The authors motivate this decision to improve the model performance. We also experimented with training models directly with negative LS and no warm-up period but found the training to become rather unstable. In some training iterations, the optimization gets stuck in a poor local minimum, substantially degrading the model's prediction accuracy. Adding the warm-up period successfully solves this problem. However, we think by selecting better warm-up parameters, the performance of models trained with negative LS can even further be improved. Due to resource restrictions, we leave more comprehensive parameter optimization for future work.
> >
> > (1) Zhang et al., The Secret Revealer: Generative Model-Inversion Attacks Against Deep Neural Networks
> > (2) Wei et al.,  To Smooth or Not? When Label Smoothing Meets Noisy Labels
> >
> > *We are currently working on adding more results for other model inversion attacks. We will add the results in the next couple of days. Please excuse the delay - getting the different code bases running and conducting the experiments takes some time. We also plan to add a user study.*

---

> > > ### Author Response · Authors · 2023-11-16
> > >
> > > Dear dNRe,
> > > we just added requested experiments and details and want to answer your remaining questions:
> > >
> > > # Model Inversion Attacks:
> > >
> > > **1.) Limited empirical study:** We focus our main analyses on the high-resolution setting since it is the most practical and realistic one. Face recognition systems using only a $64\times 64$ input image resolution are rather useless and will not find many real applications. Therefore, we think our current results focusing on the high-resolution setting are already sufficient to demonstrate the impact label smoothing can have on model inversion attacks. Our work is less about demonstrating state-of-the-art defense, but rather exploring the intriguing effect of training regularization.
> > >
> > > However, we agree that additional attack algorithms together with the evaluation of BiDO and MID in the low-resolution setting offer more insights and help to draw a complete picture. Therefore, we just added experiments on LOMMA [6] to **Table 8 in Appx. C.5**. Even if the effect is smaller than in the high-resolution space, it is clear to see that the negative label smoothing continues to attenuate the attacks.
> > >
> > > We are still working on adding experiments with VMI and PLG-MI, as requested, and also repeating all attacks against models trained with BiDO and MID. Unfortunately, the code bases are not well documented, and getting them run with our current setup takes additional time. We will add the results as soon as they become available.
> > >
> > > |Attack Type| α |↑ Acc@1 |↑ Acc@5| ↓ δKNN |↓ FID|
> > > | -------- | ------- | ------- | ------- | ------- |------- |
> > > | LOMMA (GMI) |
> > > ||0.1 |50.96% ± 3.52 |71.80% ± 4.48  |955.44 |47.44|
> > > ||0.0 | 53.64% ± 4.64| 79.60% ± 3.55 |878.36| 42.28|
> > > ||−0.05|  39.16% ± 4.25  |68.00% ± 4.49| 854.10 |39.24|
> > > |LOMMA (KED)|
> > > ||0.1  |76.52% ± 1.31 | 92.40% ± 1.05  |780.76 |33.01|
> > > ||0.0  |72.96% ± 1.29 |93.00% ± 0.83| 791.80| 33.39|
> > > ||−0.05 | 63.60% ± 1.37 |86.60% ± 0.70 | 784.43| 40.97|
> > >
> > > **2.) Unstable gradient directions due to PPA's optimization objective?** We hope that answering 1.) in our previous answer already demonstrates that the gradient stability is not due to PPA's optimization objective. To provide additional insights, we repeated the experiment but replaced the Poincaré optimization goal with a simple cross-entropy loss. We found that the general pattern is still present: the optimization on the model trained with negative label smoothing is still rather unstable, and the mean similarity between gradients is close to zero. We added **Fig. 7 in Appx. B.4.** with the results.
> > >
> > > **3.) User study**
> > > We are currently preparing the user study, but it still takes some time. We will report results as soon as they become available.

---

> > > > ### Author Response · Authors · 2023-11-16
> > > >
> > > > **4.) Compromise in model utility:** We re-trained the target models with positive LS and hard labels and stopped the training as soon as the classifier reached the test accuracy of the model trained with negative LS (see also our answer 1.) to Reviewer 22v2). As our results demonstrate, the attacks perform substantially worse on models trained with negative LS, even if all models have comparable test accuracy. For more details, we refer to **Appx. C.6.**.
> > > >
> > > > Please excuse that we also kindly disagree with the statement "Neg. LS reduces the mode accuracy by huge amounts". In the security and privacy domain, a 3 percentage points loss of accuracy is not "huge" but rather common in various defense settings. See, for example, the results in Table 1 in the BiDO paper [3]. BiDO, the current state-of-the-art defense, reduces the test accuracy by 6\% up to 33\%, and the previous approach MID also reduces the accuracy by 8\% up to 33\%. Defense mechanisms from other privacy domains, like differential privacy or membership inference attacks, also always sacrifice some performance for privacy defense. Overall, we do not think the kind of accuracy drop we observe in our paper makes the whole approach "questionable" in practice. Again, our higher motivation is not to propose a state-of-the-art defense mechanism but to showcase the impact of label smoothing regularization on a model's privacy leakage. Moreover, since we only performed a small parameter search for training with negative LS due to computational limitations, we believe that the model utility can even further be improved by selecting a better scheduler and hyperparameters.
> > > >
> > > > **5.) Standard deviation:** We computed the standard deviations for the attack accuracy analogously to [5] and [6] and added the results to **Tables 2, 3, and 8** in the Appendix. We decided to leave the standard deviations out of Table 1 to do not hurt the readability  (table 2 extends the results from Table 1, so all information is available in the paper).
> > > >
> > > > **6.) Missing related work:** Thank you for sharing those papers with us. We included them in the background and related work section.
> > > >
> > > > [3] Peng et al., Bilateral Dependency Optimization: Defending Against Model-inversion Attacks
> > > > [4] Abadi et al., Deep Learning with Differential Privacy
> > > > [5] Chen et al., Knowledge-Enriched Distributional Model Inversion Attacks
> > > > [6] Nguyen et al., Re-thinking Model Inversion Attacks Against Deep Neural Networks

---

> ### Author Response · Authors · 2023-11-20
>
> Dear dNRe,
> we just added additional model inversion experiments for PLG-MI, as well as for MID and BiDO as defense mechanisms. For BiDO and MID, we trained two models each with different defense parameter strengths. For VMI, we are also currently working on getting the code running, which is a bit tricky due to undetailed documentation and missing requirements and config files.
>
> Overall, as our results on the 7 attacks demonstrate, negative LS beats MID and BiDO as defense in most attacks (at comparable test accuracy). For PLG-MI, all defenses perform comparably. Given that the defense improvement of negative LS is even higher in the high-resolution setting of PPA, we think our empirical results have sufficiently demonstrated the defensive impact of negative LS.
>
> |  Attack Type |  Training Parameter |  ↑ Test Acc |  ↑ Acc@1 |  ↑ Acc@5 |  ↓ δKNN  | ↓ FID |
> | -------- | ------- |  ------- |  ------- |  ------- |  ------- |  ------- |
> | GMI |
> | |  Baseline α = 0.0 |  85.74% |  16.00% ± 3.75 |  36.60% ± 4.37 |  1043.22 |  52.90 |
> | |  LS α = 0.1 |  88.10% |  25.40% ± 5.06|  47.40% ± 5.09 |  1067.55 |  51.12 |
> | |  LS α = −0.05 |  80.02% |  5.92% ± 2.31 |  19.80% ± 3.91 |  1078.57 |  70.87|
> | | MID β = 0.003 |77.56% | 14.60% ± 2.80 | 30.00% ± 4.15 | 1079.17 | 56.52 |
> | | MID β = 0.01 | 67.45% | 12.56% ± 3.79 | 31.00% ± 3.69 | 1085.05 | 59.41 |
> | |  HSIC β = (0.05, 0.5) |  79.06%  |  7.40% ± 2.79 |  17.80% ± 4.55 |  1142.14 |  65.36|
> | |  HSIC β = (0.05, 1.0) |  70.18%  |  2.76% ± 1.36 | 8.60% ± 2.21 |  1227.64 |  81.60|
> | KED |
> | |  Baseline α = 0.0 |  85.74% |  43.64% ± 3.67 |  71.80% ± 3.41 |  897.54 |  42.59 |
> | |  LS α = 0.1 |  88.10%  |  68.88% ± 3.23  | 86.20% ± 2.43 | 791.11 | 24.10|
> | | LS α = −0.05 | 80.02%  | 24.10% ± 3.06  | 54.80% ± 2.88 |  953.74 |  43.56 |
> | | MID β = 0.003 | 77.56% | 60.40% ± 2.49 | 87.80% ± 2.32 | 797.99 | 27.58 |
> | | MID β = 0.01 | 67.45% | 50.44% ± 2.21 | 79.20% ± 1.80 | 821.01 | 27.02 |
> | |  HSIC β = (0.05, 0.5) | 79.06%  | 42.72% ± 4.25 | 71.60% ± 2.67 |  865.94 |  29.98|
> | |  HSIC β = (0.05, 1.0) |  70.18% |  29.00% ± 5.13  |  58.20% ± 2.60 |  932.78 |  31.95|
> |  LOMMA (GMI) |
> | |   Baseline α = 0.0 |  85.74%|   53.64% ± 4.64|   79.60% ± 3.55 |  878.36 |  42.28|
> | | LS α = 0.1 | 88.10%| 50.96% ± 3.52  | 71.80% ± 4.48 |  955.44 |  47.44|
> | | LS α = −0.05 |  80.02% |  39.16% ± 4.25 |  68.00% ± 4.49 |  854.10 |  39.24|
> | | MID β = 0.003 | 77.56% | 32.92% ± 3.59 | 57.40% ± 3.16 | 961.51 | 51.98 |
> | | MID β = 0.01 | 67.45% | 17.16% ± 3.71 | 36.80% ± 6.47 | 1044.63 | 59.07 |
> | | HSIC β = (0.05, 0.5) |  79.06%  |  47.84% ± 4.32|  74.40% ± 4.51 |  892.00 |  41.76 |
> | |  HSIC β = (0.05, 1.0) |  70.18%|  30.84% ± 4.27 |  54.20% ± 4.68 |  963.49 |  44.36|
> |  LOMMA (KED)|
> | |  Baseline α = 0.0 |  85.74% |  72.96% ± 1.29 |  93.00% ± 0.83 |  791.80 |  33.39|
> | |  LS α = 0.1 |  88.10%  |  76.52% ± 1.31 |  92.40% ± 1.05 |  780.76 |  33.01 |
> | |  LS α = −0.05 | 80.02% |   63.60% ± 1.37 |  86.60% ± 0.70 |  784.43 |  40.97 |
> | | MID β = 0.003 77.56% | 63.56% ± 1.11 | 90.60% ± 0.68 | 792.74 | 39.69 |
> | | MID β = 0.01 67.45% | 50.68% ± 1.49 | 76.60% ± 1.58 | 831.31 | 36.87 |
> | |  HSIC β = (0.05, 0.5) |  79.06% |  65.68% ± 1.41 |  86.60% ± 0.67 |  810.61 |  35.95 |
> | |  HSIC β = (0.05, 1.0) |  70.18%  |  47.64% ± 1.28 |  73.80% ± 0.94 |  860.08 |  38.77|
> | PLG-MI |
> | |  Baseline α = 0.0 |  85.74% |  71.00% ± 3.31 |  92.00% ± 3.16 |  1358.56 |  22.43|
> | | LS α = 0.1 | 88.10%|  80.00% ± 4.47 |  92.00% ± 3.16 | 1329.05|  21.89|
> | | LS α = −0.05 | 80.02%  | 72.00% ± 2.50| 89.00% ± 2.00 | 1544.82|  78.98|
> | | MID β = 0.003 | 77.56% |72.00% ± 6.08  | 89.00% ± 3.00  | 1378.50 | 20.74 |
> | | MID β = 0.01 | 67.45% | 59.00% ± 2.45 | 81.00% ± 3.87 | 1487.30 | 20.64 |
> | | HSIC β = (0.05, 0.5) | 79.06%  | 70.00% ± 4.00  | 85.00% ± 3.16 | 1433.48 | 25.37|
> | | HSIC β = (0.05, 1.0) | 70.18%  | 42.00% ± 4.80 |  62.00% ± 5.10 | 1585.35|  30.52|
> | RLB-MI|
> | | Baseline α = 0.0 | 85.74%|  52.00% | 75.00%|  - | -|
> | | LS α = 0.1|  88.10%|  65.00% |  84.00% | - | -|
> | | LS α = −0.05 | 80.02%  | 19.00%|  48.00%|  -|  -|
> | | MID β = 0.003 | 77.56% | 27.00% | 41.00% | - | - |
> | | MID β = 0.01 | 67.45% | 20.00% | 42.00% | - | - |
> | | HSIC β = (0.05, 0.5) | 79.06% |  35.00% | 57.00%| -| -|
> | | HSIC β = (0.05, 1.0)|  70.18% | 25.00% | 49.00%| -| -|
> | BREP-MI|
> | |  Baseline α = 0.0 |  85.74% |  49.00%|  73.67%|  -|  -|
> | | LS α = 0.1 | 88.10%  | 56.33%  | 77.00% | -|  -|
> | | LS α = −0.05|  80.02% |  48.70% |  70.50%|  - | -|
> | | MID β = 0.003 | 77.56% | 53.43% | 76.05% | - | - |
> | | MID β = 0.01 | 67.45%| 43.17% | 69.25% | - | - |
> | | HSIC β = (0.05, 0.5) | 79.06% |  42.60% | 66.90%| -| -|
> | | HSIC β = (0.05, 1.0) | 70.18%  | 27.10%|  50.30%| -| -|
>
> For more details, please visit **Appx. C.5.** of the paper.

---

> > ### Author Response · Authors · 2023-11-20
> >
> > As requested, we conducted a qualitative user study to compare the effects of different defense methods on the visual similarity between attack results and the true identity. We compared the attack results on the ResNet-152 FaceScrub and CelebA models trained with negative LS to the attack results against models trained with MID and BiDO, respectively. The study was conducted using the manual labeling service of https://thehive.ai/. More specifically, we randomly sampled $150$ target classes from the FaceScrub and CelebA datasets, respectively. We then took for each target model and target class the attack results that achieved the highest robustness on the target model using PPA's final selection algorithm. For each class, the annotators were presented with $5$ randomly selected images of the true identity and the $3$ attack results from the attack against negative LS, MID, and BiDO models. The annotators were then asked to "select the person (options 1-3) that looks the least similar to the person depicted in the example images". Noticeably, the annotators were not provided any information on the different defense mechanisms or even the attack setting at all to avoid undesired side effects during the labeling process. Each sample was labeled by three different annotators and the final decision was made on a majority vote. In total, 37 annotators took part in the labeling process. To filter out unreliable annotators, some honeypot tasks were added to ensure the annotation quality.
> >
> > The following table states the user study results as the relative preference for the individual defense mechanisms. For the FaceScrub model, the annotators found about 48% of the attack results from attacks against the negative LS model to look the least similar to the target class. For the CelebA experiment, even 71.5% of the results were preferred. Overall, this means that the defensive effect of negative LS was qualitatively higher valued than MID and BiDO as defense mechanisms, particularly on the CelebA models. At the same time, negative LS also maintained the model's utility in terms of test accuracy better than the other defense mechanisms.
> >
> > |Model |Test Acc | Acc@1| δface |↑↑↑ User Preference | Test Acc | Acc@1 | δface|  User Preference|
> > | -------- | ------- | ------- | ------- | ------- | ------- | ------- | ------- | ------- |
> > FaceScrub ||||| CelebA|||
> > |Neg. LS |91.5% |14.3%  | 1.23  |**47.76%** | 83.6% |26.4% |1.04| **71.53%**|
> > |MID |91.1% |92.0% |0.72  | **17.16%** | 80.4% |78.0% | 0.74 | **10.22%**|
> > |BiDO| 87.1% |45.4% | 0.91| **35.07%** |79.9%| 63.7% | 0.81| **18.25%**|
> >
> > Fore more details, please take a look at **Appx. C.10** in the paper.

---

> > ### Author Response · Authors · 2023-11-21
> >
> > Dear dNRe,
> > we just added the MID results to the table above. We hope with this we answered all your questions and added all requested experimental results. If you have any questions left, we are happy to answer them before the end of the discussion phase.
> >
> > Best,
> > the authors

---

> ### Comment · Reviewer_dNRe · 2023-11-22
> **Reviewer Response**
>
> Thank you authors for the extensive rebuttal. I appreciate the detailed response, additional results and user studies. I also like to remark that this paper is entirely empirical.
>
> After reading the rebuttal and the other reviews, below I include my response for each portion:
>
> 1. **User Study**: The human preference results presented are supportive and provide valuable insights. However, I suggest a potential refinement to the user study's design for future work. The central question of the study could be more effectively framed as “Which image resembles the person above?” This approach would directly assess the comparative effectiveness of BiDO/MID against your method. Such a framing could offer a more objective understanding of the efficacy of negative label smoothing. Nonetheless, I acknowledge that the current results are satisfactory.
>
> $~$
>
> 2. **Figure 4, column 3**. I’m still confused by the penultimate visualization of Figure 4, column 3. In fact, if the classifier accuracy =91.45% for the model with $\alpha=-0.05$, why are the penultimate representations of different classes mixed up? I.e.: Did you project the representations before dimensionally reducing them similar to [A, B] prior to plotting.
>
> $~$
>
> 3. **Contradictory rebuttal results reported for LOMMA and PLG-MI.** The rebuttal results reported in Table 8 actually do not support the thesis of the paper:
> $~$
> - Why does positive label smoothing provide better defense compared to no label smoothing for LOMMA (GMI), despite having higher test accuracy.. I copy the reported rebuttal results below:
>
> | LOMMA (GMI)              | Test Acc | $\alpha$ | Top1 Attack Acc |
> | ------------------------ | -------- | --------- | ----------------- |
> | Baseline                 | 85.74%   | 0.0         | 53.64% ± 4.64     |
> | Positive Label Smoothing | 88.10%   | 0.1       | 50.96% ± 3.52     |
>
> $~$
> - Why does negative label smoothing provide worse defense compared to no label smoothing for PLG-MI, despite a significantly lower test accuracy? I copy the reported rebuttal results below:
>
> | PLG-MI                   | Test Acc | $\alpha$ | Top1 Attack Acc |
> | ------------------------ | -------- | --------- | --------------- |
> | Baseline                 | 85.74%   | 0.0         | 71.00% ± 3.31   |
> | Negative Label Smoothing | 80.02%   | \-0.05    | 72.00% ± 2.50   |
>
>
> **Given that this paper is entirely empirical, critical results using state-of-the-art model inversion algorithms do not support this paper’s thesis.**
>
> $~$
>
> 4. **Lack of Theoretical/Analytical Support.** A major limitation of this work is the absence of solid theoretical/ analytical backing for the claims made. Beyond the toy experiment illustration in Section 3, the rationale behind how positive and negative label smoothing impacts model inversion attacks as suggested in the paper remains unclear. This concern is further compounded by the contradictory results for PLG-MI and LOMMA (GMI) in the rebuttal, which further obscures the paper's claims.
>
>
> $~$
>
>
> **Therefore, while I sincerely appreciate the authors' comprehensive rebuttal and the efforts put into it, I find myself still hesitant to vote for the acceptance of this paper in its current form (I have increased my rating from 3->5 as the rebuttal has addressed some of my concerns).**

---

> > ### Author Response · Authors · 2023-11-23
> >
> > Dear Reviewer dNRe,
> >
> > thank you very much for acknowledging our rebuttal and increasing your score.
> >
> > We are still eager to address your concerns and convince you of our work. Due to the close end of the rebuttal phase, please excuse that our answers and additional results are limited.
> >
> > **1.) User study**: we thank you for suggesting improvements for the user study. If you desire, we can add a second study to the final version of the paper, if it gets accepted.
> >
> > **2.) Fig. 4** We did not use the weight-based projection as suggested by [A, B] but rather directly projected the penultimate activations into 2D space by using t-SNE. We will look into the visualization of Müller et al. but will probably not finish it until the rebuttal deadline.
> >
> > 3.) **Lack of Theoretical/Analytical Support**: Providing a grounded theoretical framework for our work already requires a universally accepted theoretical framework for model inversion attacks. Unfortunately, to date, there is no such framework available, and since sensitive features and privacy leakage themselves are hard to clearly define, this is no trivial task. We also stress that previous attack and defense methods all also not provide a theoretical framework. However, we believe that our extensive experiments and analyses already provide proof for our statements.
> >
> > **4.) Why does positive label smoothing provide better defense compared to no label smoothing for LOMMA (GMI), despite having higher test accuracy?** Please note that positive LS smoothing increases the attack success/privacy leakage in 7 out of 8 attack settings: PPA (high resolution), GMI, KED, KED + LOMMA, PLG-MI, RLB-MI, and BREP-MI. So there is a clear pattern. For GMI + LOMMA we think there are two reasons why positive LS does not outperform the hard-label case. First, positive label smoothing might hinder the knowledge distillation process by training the augmentation models, as previously discussed, e.g., by Müller et al. [A]. So including the augmentation models in this attack setting might actually limit the attack's success. KED+LOMMA uses the same augmentation models but also includes the target-specific GAN, which provides a substantial attack performance boost against the positive LS model, as demonstrated by the attack results for KED without LOMMA. Second, the target objective of LOMMA (logit score + regularization) might interfere with the fact that label smoothing encourages representations in the penultimate layer to group in tight equally distant clusters, see also [A]. So we believe that the overall reason why GMI+LOMMA does not perform better on the positive LS models probably is due to the attack design that does not work well with the model's feature representations.
> >
> > **5.) Why does negative label smoothing provide worse defense compared to no label smoothing for PLG-MI, despite a significantly lower test accuracy?** Indeed, the attack accuracy indicates misleading attack results. Whereas the attack accuracy is quite similar for both models, the other attack metrics (feature distance and FID score) are substantially worse for the negative LS model. Since the attack accuracy is susceptible to adversarial examples and does not make a statement about the sample quality, it is important to also consider the other metrics.
> >
> > |Attack Type|	Training Parameter|	↑ Test Acc	|↑ Acc@1|	↑ Acc@5|	↓ δKNN	|↓ FID|
> > |----------|---------------|---------------|--------------|------------|-----------------|-----------------|
> > | Baseline |α = 0.0|	85.74%	|71.00% ± 3.31	|92.00% ± 3.16|	1358.56	|22.43|
> > |LS| α = −0.05	|80.02%	|72.00% ± 2.50|	89.00% ± 2.00	|1544.82|	78.98|
> >
> > To provide additional qualitative proof of the defensive effect of negative LS, we just plotted attack results for the first ten classes in **Fig. 11** in Appx. C.5. Importantly, the samples were not cherry-picked but we took the first generated sample for each class. When taking a look at the image it becomes clear that PLG-MI indeed produces qualitative worse results for the negative LS model. So negative LS in this case limits the privacy leakage by limitating the constructed samples' quality. Also, the variance of the different samples of one class is very low (not depicted in the image) whereas the attack against the other models constructs visually different samples for each single class.
> >
> > **So we stress that the results for LOMMA (GMI) and PLG-MI are not contradictory to the general message of the paper ("Label Smoothing Can Be a Privacy Shield but Also a Catalyst for Model Inversion Attacks"). For LOMMA (GMI), the attack design simply does not match the characteristics of positive label smoothing training, and for PLG-MI the qualitative samples and other metrics clearly show that negative LS indeed mitigates the privacy leakage of the model.**
> >
> > Again, thank you very much for your insightful review. We are happy to make any suggested changes and additional results to the final paper.
> >
> > Best,
> > the authors

---

### Official Review · Reviewer_22v2 · 2023-11-06

**Soundness:** 2 fair
**Presentation:** 4 excellent
**Contribution:** 3 good
**Rating:** 6
**Confidence:** 2

**Summary:**

The paper shows that model trained with label smoothing can be more vulnerable to model inversion attacks while negative label smoothing can be a defense for the attacks.

**Strengths:**

The paper looks at an interesting and important problem of how model training can affect model’s vulnerability to MIAs.
The experimental results in general justifies the statements well.
The presentation is clear.

**Weaknesses:**

I’m still wondering how much we can conclude the relation between vulnerability of MIAs and training methods (normal / smoothing / negative smoothing) from the empirical results shown here. For example,

- different training methods lead to different model test accuracy (Table 1) especially for CelebA. I think it’s fairer to compare the attack accuracy under the same model accuracy (e.g. early stop the label-smoothed model at a lower accuracy iteration), as it might be normal for a higher accuracy model leak more. It might also be interesting to look at training accuracy of the models to understand how well they generalize.

- we’re looking at one particular attack algorithm here. I think it’s natural to ask whether another algorithm, or maybe an adjusted version this this algorithm can achieve a different result. For example, if the attacker knows that a model is trained with negative label smoothing (and thus has a different calibration than a normally-trained model), can they possibly sample more initial latent embeddings in the first stage, or adjust their objective function to incorporate with the calibration of this model in the second stage?

**Questions:**

In Fig 6b, why is the gradient similarity for label smoothing lower & with higher variance than that of hard label? I must admit that I don’t have a good intuition here but I was kind of expecting the lines of smoothing and negative smoothing to stay on two sides of the hard label’s line.

(And those mentioned above.)

---

> ### Author Response · Authors · 2023-11-15
>
> Dear Reviewer 22v2,
> thank you very much for reviewing our paper. We are glad that you find our paper interesting and important. We will try to answer all your questions sufficiently.
>
> **1.) Attack comparison under same model accuracy:** We re-trained the models with positive LS and standard cross-entropy loss and stopped each training run as soon as the model's prediction accuracy on the holdout test set matched that of the model trained with negative LS. We then repeated the attacks against these models, which all achieve comparable prediction accuracy, to see if the attack success against these models drops. Our results clearly demonstrate that even under comparable test accuracy, the attacks still perform well and substantially better than on the negative LS models. This indicates that reducing a model's prediction accuracy is not the driving factor behind negative LS as a defense mechanism. We note that for FaceScrub, the attack against the model trained with positive LS performs slightly worse than for the model trained without any LS. However, the positive LS model already achieves the target accuracy after only 4 epochs, while the model without LS was trained for 19 epochs. Consequently, the positive LS model has seen much fewer samples and underwent fewer parameter updates to incorporate individual class features.
>
> |  FaceScrub |  α |  ↑ Test Acc |  ↑ Acc@1 | ↑ Acc@5 | ↓ δface | ↓ δeval |  ↓FID |
> | -------- | ------- |------- |------- |------- |------- |------- |------- |
> No Label Smoothing |0.0 |91.74%| 89.68%| 99.00% |0.7382 |133.12 |50.43|
> Positive Label Smoothing| 0.1 |92.13%| 87.16% |97.20% |0.7766 |135.73| 49.56|
> Negative Label Smoothing |−0.05| 91.45% |14.34%| 30.94% |1.2320 |239.02 |59.38 |16.45% |15.73%|
>
> |  CelebA |  α |  ↑ Test Acc |  ↑ Acc@1 | ↑ Acc@5 | ↓ δface | ↓ δeval |  ↓FID |
> | -------- | ------- |------- |------- |------- |------- |------- |------- |
> |No Label Smoothing |0.0 |84.02%| 69.36% |90.60%| 0.7899| 344.82| 49.60|
> |Positive Label Smoothing |0.1 |86.78% |76.64%| 92.84%| 0.785| 332.53| 50.05|
> |Negative Label Smoothing |−0.05 |83.59%| 26.41%| 49.96%| 1.0420| 441.67 |61.30 |7.08%| 5.89%|
>
> Our results including a more detailed experimental setting and more elaborated discussion are also stated in **Appendix C.6**.

---

> > ### Author Response · Authors · 2023-11-15
> >
> > **2.) Can an adjusted attack algorithm achieve different results?** We conducted additional experiments by changing or replacing specific aspects of PPA. Specifically, we increased the number of initial sample vectors by factor 5, we experimented with different learning rates, we added an additional prior loss based on the GANs discriminator, and we tested two different loss functions from recent literature [1, 2]. While some of the changes could indeed improve the attack's success, the overall attack performance is still substantially below those for models trained without negative LS. For example, the FaceNet distance, which states the visual similarity between reconstructed samples and training samples from the target class, is still about 0.3-0.7 higher than for attacks against the model trained without any LS (avg. FaceNet distance 0.71). We, therefore, conclude that by selecting good attack parameters and making adjustments to the optimization goal, one might improve the attack success but compared to models trained without LS or with positive LS, the attack still falls short by a large margin.
> >
> > | Variant |↑ Acc@1 |↑ Acc@5 |↓ δface |↓ δeval |↓FID |
> > | -------- | ------- |------- |------- |------- |------- |
> > Default |  14.34% |  30.94% |  1.2320 |  239.02 |  59.38 |  16.45% |  15.73% |
> > Increased Latent Vector Sampling |  19.24% |  39.64% |  1.178 |  225.39 |  63.87 |
> > Lower Learning Rate |  2.72% |  9.40%  |  1.381 |  222.13 |  75.94  |
> > Higher Learning Rate |  19.44%|  36.60%|  1.258 | 1031.79 | 86.44 |
> > Cross-Entropy Loss (Zhang et al., 2020) | 23.56% | 55.92%|  1.01 | 156.79 | 62.39|
> > Prior Loss (Zhang et al., 2020)|  12.04% | 29.60% | 1.224|  218.58|  64.83 |
> > Identity Loss (Nguyen et al., 2023) | 22.84%|  40.56% | 1.197 | 186.52 | 84.48 |
> >
> > Our results including a more detailed experimental setting and more elaborated discussion are also added in **Appendix C.7**.
> >
> > (1) Zhang et al., The Secret Revealer: Generative Model-Inversion Attacks Against Deep Neural Networks
> > (2) Nguyen et al., Re-Thinking Model Inversion Attacks Against Deep Neural Networks
> >
> > **3.) Why is the gradient similarity for positive LS lower than for training with hard labels?** During the attack optimization process, the attack explores the GAN's latent space by guidance through the target model. We hypothesize that the optimization landscape defined by the target model trained with LS - compared to the model trained with hard labels - leads to more pronounced changes in direction since the model reveals more fine-grained details on the features. Recovering such fine-grained features might need more adjustments to the optimization direction during the optimization since the next gradient step is based on the current position in the latent space. Keeping the same direction might probably is not ideal for recovering certain features. So we suppose that the model trained with hard labels provides not enough guidance signal but rather keeps the overall optimization direction, which leads to generally good results but some fine-grained details might not be recovered. The model trained with positive LS, however, provides a stronger guidance signal, which leads to locally more pronounced changes in the optimization direction. The model trained with negative LS often fails to exploit local information sufficiently and provide a stable guidance signal.

---

### Author Response · Authors · 2023-11-21
**Overall Response**

We sincerely thank all reviewers for their valuable time and effort in reviewing our work. We appreciate the reviewers' comments and recognition of our work.

We hope that we have sufficiently answered all comments and questions with our detailed response. We have highlighted all additional content in the paper with blue color. In particular, we have added/updated the following contents in the paper:
- As requested by reviewer **dNRe**, we have added additional attack results, including LOMMA and PLG-MI, in Appendix C.5 to highlight the impact of regularizing label smoothing on attack success. We have also added more results for models trained with MID and BiDO defenses to show that in most cases, training with negative label smoothing provides a better tradeoff between utility and privacy than the two existing defenses.
- Because reviewers **22v2, dNRe, and wJ7G** wondered whether the defensive effect of negative labeling smoothing was due to reduced model utility, we added experiments in Appendix C.6 showing that even with comparable prediction accuracy, the privacy leakage is substantially lower for models with negative labeling smoothing.
- As requested by reviewer **22v2**, we have added an analysis on possible attack improvements for PPA in Appendix C.7 to show that negative label smoothing reduces the loss of privacy even when different optimization criteria and hyperparameters are chosen.
- As requested by reviewer **dNRe**, we conducted a user study to evaluate whether negative label smoothing indeed better restricts the privacy leakage of a model compared to other defensive measures. As the results in Appendix C.10 show, the defensive effect of negative label smoothing is favored by participants over MID and BiDO in both datasets studied.
- As requested by the reviewers **uLpV, wJ7G and EAeB**, we also conducted experiments to test whether training with positive or negative label smoothing affects the robustness of a model against adversarial examples or backdoor attacks. As our results in Appendices C.8 and C.9 show, negative label smoothing does not have a negative impact on the robustness of a model, but rather can make it more robust against these types of attacks.

Please notify us if you have any further suggestions. We would be glad to discuss and answer them.

Thank you very much.

Yours sincerely,
the Authors

---

### Meta-Review · Area_Chair_e7Vt · 2023-12-06

**Metareview:**

The paper studies the impact of label smoothing (LS) in terms of privacy. In LS (Szegedy et al., 2016), instead of using the indicator vector of the label when computing the loss in training, we take an $(1 - \\alpha, \\alpha / C)$ linear combination between the indicator vector and the all-one vector (where $\\alpha > 0$ and $C$ denote the number of labels). This has been shown to increase utility of the model. This paper shows however that this also leads to higher privacy leakage, in particular model inversion attack (MIA). In MIA (Fredrickson et al., 2015), an adversary gets to see the model and tries to reconstruct some datapoints in the dataset. The paper shows that the SOTA MIA (Struppek et al., 2022) can construct datapoints much more accurately when the model is trained with LS compared to without (even for similar model utility). The authors then propose a fix to this: use *negative* LS i.e. setting $\\alpha < 0$. The authors show empirically that this significantly decreases the accuracy of the attack compared to models trained without LS or with positive (usual) LS .

The authors also provide a high-level explanation on this phenomenon. Typical approach for MIA is to find a point that has very high confidence in one class using standard optimization techniques like gradient descent. Positive LS makes the confidence vector more smooth so such optimization is more successful. Meanwhile, negative LS prefers a more flat confidence vector region within each class (as soon as we cross the decision boundary, the confidence vector becomes close to one quickly) since it penalizes wrong prediction even more. This makes the MIAs fail.

## Strengths

- LS is a popular technique and no privacy impact has been investigated. This paper clearly demonstrates that (positive) LS results in more vulnerability against MIA, which is very concerning.

- To me, the finding is very surprising. Positive LS is the same as the expected loss if we were to apply label-differential privacy protection by applying randomized response to the labels. So I had always been expecting that positive LS would provide a better privacy guarantee (compared to no LS). But the finding is completely the reverse.

- The proposed negative LS is interesting and seems to work well based on empirical results of the paper.

- In addition to empirical results, the high-level examples that the authors provide are illustrative

## Weaknesses

- There is no theoretical analysis for the protection provided by negative LS. Due to this, it is unclear whether the phenomenon occurs only because of the current attack approaches or whether one will be able to come up with a smarter attack that bypasses negative LS.

**Justification For Why Not Higher Score:**

As stated above, the lack of theoretical guarantees prevent this from being a spotlight/oral paper for me.

**Justification For Why Not Lower Score:**

This paper provides a surprising insight into how a popular technique affects privacy leakage with comprehensive experiments and clear high-level explanations. Thus, I think it clearly passes the bar for ICLR.

---

### Decision · Program_Chairs · 2024-01-16

Accept (poster)